# Hidden chemical order in disordered Ba₇Nb₄MoO₂₀ revealed by resonant X-ray diffraction and solid-state NMR

Yuta Yasui[1], Masataka Tansho[2], Kotaro Fujii [ORCID][1], Yuichi Sakuda[1], Atsushi Goto [ORCID][2], Shinobu Ohki[2], Yuuki Mogami[2], Takahiro Iijima[3], Shintaro Kobayashi [ORCID][4], Shogo Kawaguchi [ORCID][4], Keiichi Osaka[5], Kazutaka Ikeda [ORCID][6,7,8], Toshiya Otomo [ORCID][6,7,8,9] & Masatomo Yashima [ORCID][1] ✉

The chemical order and disorder of solids have a decisive influence on the material properties. There are numerous materials exhibiting chemical order/disorder of atoms with similar X-ray atomic scattering factors and similar neutron scattering lengths. It is difficult to investigate such order/disorder hidden in the data obtained from conventional diffraction methods. Herein, we quantitatively determined the Mo/Nb order in the high ion conductor Ba₇Nb₄MoO₂₀ by a technique combining resonant X-ray diffraction, solid-state nuclear magnetic resonance (NMR) and first-principle calculations. NMR provided direct evidence that Mo atoms occupy only the $M2$ site near the intrinsically oxygen-deficient ion-conducting layer. Resonant X-ray diffraction determined the occupancy factors of Mo atoms at the $M2$ and other sites to be 0.50 and 0.00, respectively. These findings provide a basis for the development of ion conductors. This combined technique would open a new avenue for in-depth investigation of the hidden chemical order/disorder in materials.

Structural order and disorder have attracted considerable attention because of their correlation with material properties[1-16]. Chemical (occupational) order and disorder have been studied mainly by crystal structure analysis using diffraction data. Such order and disorder are often observed among elements demonstrating similar X-ray atomic scattering factors and similar neutron scattering lengths. Here, we consider the chemical order between two elements $X$ and $Y$ ($X/Y$ order) and define the Scattering Contrast Score of elements $X$ and $Y$, SCS($X$, $Y$) as a measure of the contrasts in X-ray and neutron scattering powers

between the $X$ and $Y$ elements.

$$\text{SCS}(X,Y) = \left| \frac{N(X) - N(Y)}{N(X) + N(Y)} \right| + \left| \frac{\text{Re}[b(X)] - \text{Re}[b(Y)]}{\text{Re}[b(X)] + \text{Re}[b(Y)]} \right| \quad (1)$$

Here $N(X)$ and $\text{Re}[b(X)]$ are the number of electrons and real part of the coherent neutron scattering length $b$ of atom $X$, respectively. There are numerous pairs of $X$ and $Y$ elements with low SCS values (ex. ~300 $X/Y$ pairs with SCS lower than 0.15; red parts in Fig. 1). However, it is

[1]Department of Chemistry, School of Science, Tokyo Institute of Technology, 2-12-1-W4-17, O-okayama, Meguro-ku, Tokyo 152-8551, Japan. [2]NMR Station, National Institute for Materials Science (NIMS), 3-13 Sakura, Tsukuba, Ibaraki 305-0003, Japan. [3]Institute of Arts and Sciences, Yamagata University, 1-4-12 Kojirakawa-machi, Yamagata, Yamagata 990-8560, Japan. [4]Diffraction and Scattering Division, Japan Synchrotron Radiation Research Institute (JASRI), SPring-8, 1-1-1 Kouto, Sayo-cho, Sayo-gun, Hyogo 679-5198, Japan. [5]Industrial Application and Partnership Division, Japan Synchrotron Radiation Research Institute (JASRI), SPring-8, 1-1-1 Kouto, Sayo-cho, Sayo-gun, Hyogo 679-5198, Japan. [6]Institute of Materials Structure Science, High Energy Accelerator Research Organization (KEK), 203-1 Shirakata, Tokai, Ibaraki 319-1106, Japan. [7]J-PARC Center, High Energy Accelerator Research Organization (KEK), 2-4 Shirakata-Shirane, Tokai, Ibaraki 319-1106, Japan. [8]School of High Energy Accelerator Science, The Graduate University for Advanced Studies, 203-1 Shirakata, Tokai, Ibaraki 319-1106, Japan. [9]Graduate School of Science and Engineering, Ibaraki University, 162-1 Shirakata, Tokai, Ibaraki 319-1106, Japan. ✉e-mail: yashima@cms.titech.ac.jp

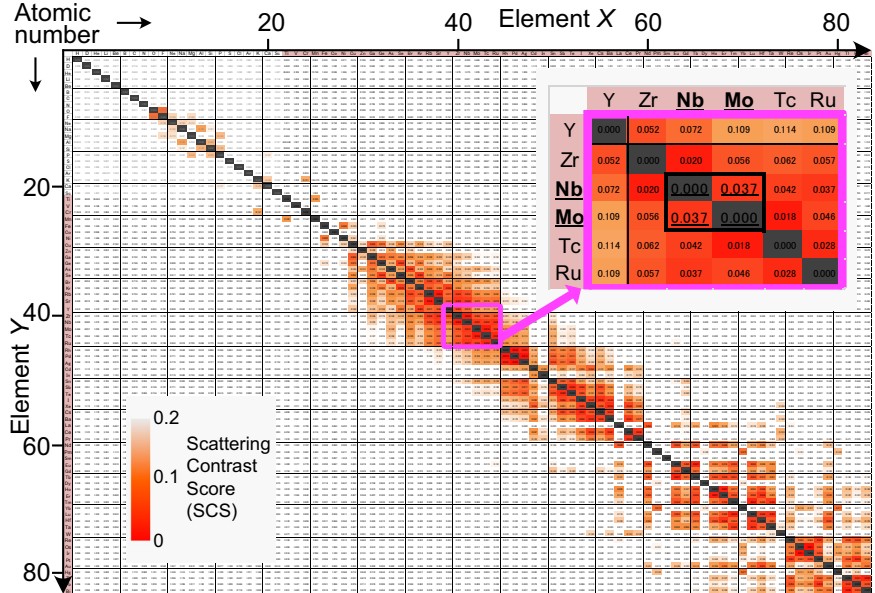

**Fig. 1 | Numerous pairs of X and Y elements having low scattering contrast score SCS(X, Y).** Each number stands for the SCS(X, Y) value. Neutron scattering lengths are taken from the NIST website[68]. All the SCS(X, Y) values are available in Supplementary Data 1 and Supplementary Table 16.

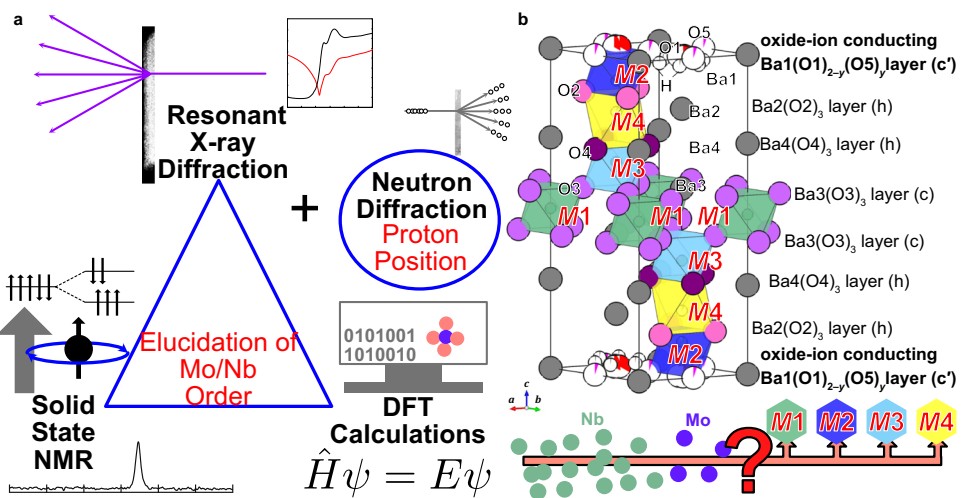

**Fig. 2 | Strategies for the elucidation of the Mo/Nb order and complete crystal structure of Ba₇Nb₄MoO₂₀·0.15 H₂O. a** Combined technique to determine the crystal structure and Mo/Nb order of Ba₇Nb₄MoO₂₀·0.15 H₂O. **b** Refined crystal structure showing the $M$1, $M$2, $M$3 and $M$4 sites of Mo and Nb atoms in Ba₇Nb₄MoO₂₀·0.15 H₂O where the Mo and Nb atoms are assumed to be completely disordered.

difficult to investigate the $X/Y$ chemical order hidden in conventional X-ray and neutron diffraction. Thus, the chemical order is an important unresolved issue with numerous materials (Supplementary Table 1). Herein, we propose a technique to elucidate the chemical order, which is a combination of resonant X-ray powder diffraction (RXRD)[17–21] and solid-state nuclear magnetic resonance (NMR)[22–25] assisted by density functional theory (DFT) calculations[26–33]. Most materials are polycrystalline or powdered. In contrast to single-crystal X-ray and neutron diffraction, this combined technique can be widely applied to both polycrystalline and powdered samples. Direct evidence of the chemical order can be obtained by NMR[23]; however, it is difficult to quantitatively analyse the chemical order among the constituent elements. In contrast, RXRD enables the quantitative determination of the chemical order by the refinement of occupancy factors, although the refinement results using powder diffraction data are often dependent on the initial structural model. A reliable, quantitative chemical order can be obtained by the present combined technique of

NMR and RXRD. We call this combined technique as RXRD/NMR method.

In this study, we aim to elucidate the Mo/Nb order/disorder in a high ion conductor Ba₇Nb₄MoO₂₀·0.15 H₂O using the RXRD/NMR method (Fig. 2a). We chose Ba₇Nb₄MoO₂₀·0.15 H₂O, because Ba₇Nb₄MoO₂₀-based oxides and related compounds are emerging materials with high ion conduction, structural disorder and high chemical stability[11,34–40]. The crystal structures of Ba₇Nb₄MoO₂₀-based oxides have been extensively investigated. However, all the structural refinements were performed assuming the complete Mo/Nb disorder[11,34–36,38,39], because the Mo⁶⁺ and Nb⁵⁺ cations have both (i) the same number of electrons leading to almost the same X-ray atomic scattering factors and (ii) almost the same neutron scattering lengths (6.715 and 7.054 fm for Mo and Nb, respectively). This indicates a small SCS value for the Mo/Nb pair of 0.037. Because the ions migrate in the oxygen-deficient c′ layers of Ba₇Nb₄MoO₂₀-based oxides[34,36,38,39], the determination of the chemical order/disorder of Mo and Nb atoms at

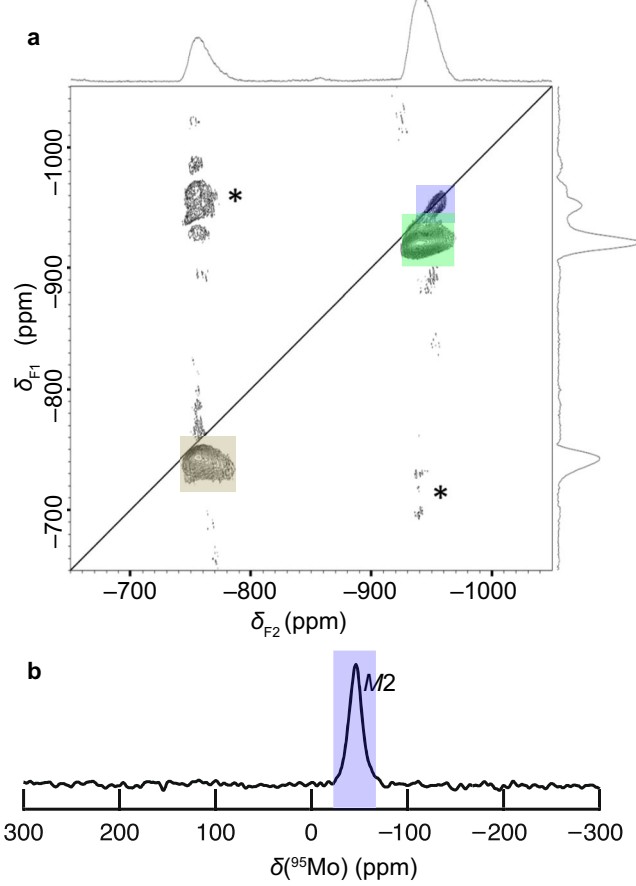

**a**

**b**

*M2*

**Fig. 3 | Solid-state NMR spectra of Ba₇Nb₄MoO₂₀·0.15 H₂O, showing the site assignment and Mo order. a** 2D ⁹³Nb 3QMAS NMR spectrum and **b** 1D ⁹⁵Mo MAS NMR spectrum of Ba₇Nb₄MoO₂₀·0.15 H₂O. An asterisk * denotes spinning sidebands.

preliminary Rietveld analyses of Ba₇Nb₄MoO₂₀·0.15 H₂O were performed using neutron diffraction (ND) data and conventional synchrotron X-ray diffraction (SXRD) data recorded with 0.6994806(5) Å X-ray far from the Nb *K*-edge, based on the Mo/Nb disordered model (Supplementary Note 1 and Supplementary Fig. 2 for details). Ba₇Nb₄MoO₂₀·0.15 H₂O was confirmed to be a *P3̄m1* hexagonal perovskite polytype 7H with four Mo/Nb cation sites (*M*1, *M*2, *M*3 and *M*4) (Fig. 2b). The occupancy factors were determined as follows:

$$g(Nb; Mi) + g(Mo; Mi) = g(Ba; Baj) = g(O; Ok) = 1,$$
$$g(Nb; M2) + g(Mo; M2) = 0.92,$$
$$g(Nb; M4) + g(Mo; M4) = 0.08,$$
$$(i = 1 \text{ and } 3; j = 1, 2, 3, \text{ and } 4; k = 2, 3, \text{ and } 4)$$

(2)

Here, $g(Nb; Mi) + g(Mo; Mi) = g(Nb_{0.8}Mo_{0.2}; Mi)$, and the $g(X; s)$ denotes the occupancy factor of $X$ atoms at $s$ site. The refined crystal parameters of Ba₇Nb₄MoO₂₀·0.15 H₂O were consistent with those reported in the literature[11,38].

**Direct experimental evidence for Mo order at *M*2 site by NMR**
We performed ⁹³Nb and ⁹⁵Mo solid-state NMR experiments on Ba₇Nb₄MoO₂₀·0.15 H₂O at a high magnetic field (18.8 T), which enabled the selective observation of Nb and Mo cations, respectively[23]. Figure 3a and Supplementary Fig. 3 show two-dimensional (2D) ⁹³Nb triple-quantum magic angle spinning (3QMAS) and one-dimensional (1D) ⁹³Nb magic angle spinning (MAS) NMR spectra, respectively. Three peaks are observed in each ⁹³Nb spectrum, indicating the presence of three Nb sites in Ba₇Nb₄MoO₂₀·0.15 H₂O. In contrast, in the 1D ⁹⁵Mo MAS NMR spectrum, only one peak is observed (Fig. 3b), indicating a single Mo site and Mo order in Ba₇Nb₄MoO₂₀·0.15 H₂O.

To assign the NMR peaks to different crystallographic sites, we performed gauge-including projector augmented wave (GIPAW) DFT calculations of NMR parameters[26–29] with the VASP programme[41]. To validate this method, the calculated ⁹³Nb and ⁹⁵Mo NMR parameters were computed for 13 niobates and 11 molybdates (Supplementary Tables 2, 3). The experimental and calculated ⁹³Nb and ⁹⁵Mo NMR parameters show good correlations (Supplementary Fig. 5). Thus, we can assign the Nb and Mo peaks by comparing the experimental and calculated NMR parameters of Ba₇Nb₄MoO₂₀·0.15 H₂O. For this purpose, the atomic positions in ten possible structural models with different Nb and Mo configurations were optimised by DFT calculations with the *P*1 space group (Supplementary Figs. 6, 7). The NMR parameters of the optimised structures were estimated by the GIPAW DFT calculations. The calculated peak positions for (Mo2)O₄ tetrahedron of Ba₇Nb₄MoO₂₀ ranged from −29 to −36 ppm depending on the structural model, which is close to the experimental peak position of −47 ppm for Ba₇Nb₄MoO₂₀·0.15 H₂O. (Table 1), where Mo2 is the Mo atom at the *M*2 site. The calculated quadrupolar coupling constant |$C_Q$| values ranged from 0.36 to 0.90 depending on the structural model, which is consistent with the observed value (≤ 2 MHz). In contrast, the peaks calculated for

the crystallographic *M*2 site near the c′ layer is essential (Fig. 2b). Thus, the chemical order of Mo and Nb atoms at the *M*2 site is an important unsolved issue. Herein, we report the chemical order of Mo atoms at the *M*2 site near the c′ layer, which offers unprecedented insight into the understanding of the ion diffusion mechanism in hexagonal perovskite-related oxides.

## Results
A single hexagonal phase of Ba₇Nb₄MoO₂₀·0.15 H₂O was prepared by the solid-state reactions (Supplementary Fig. 1). The lattice parameters of Ba₇Nb₄MoO₂₀·0.15 H₂O were determined to be $a = 5.8654(3)$ and $c = 16.5390(3)$ Å using the X-ray diffraction data of the mixture of Ba₇Nb₄MoO₂₀·0.15 H₂O sample and internal standard silicon. To determine the occupancy factors of Nb₀.₈Mo₀.₂, Ba and O atoms,

**Table 1 | Calculated and experimental ⁹⁵Mo NMR parameters of Ba₇Nb₄MoO₂₀, showing the peak assignment to the *M*2 site**

| Models ᵃ | Site | Polyhedron ᵇ | DFT-calculated peak position (ppm)ᶜ | DFT-calculated \|$C_Q$\| (MHz)ᵈ | Experimental peak position (ppm) | Experimental \|$C_Q$\| (MHz)ᵈ |
|---|---|---|---|---|---|---|
| (4), (5) | M1 | (Mo1)O₆ | +269 ~ +270 | 1.47, 2.00 | not observed | |
| (1)–(3), (5) | M2 | (Mo2)O₄ | −29.1 ~ −36.8 | 0.36 ~ 0.90 | −47 | ≤ 2 |
| (7)–(10) | M2 | (Mo2)O₅ | −57 ~ +80 | 5.1 ~ 10.7 | not observed | |
| (6) | M3 | (Mo3)O₆ | +145 | 2.87 | not observed | |
| (7), (8) | M4 | (Mo4)O₆ | +47, +49 | 5.17, 5.23 | not observed | |

ᵃEach model is shown in Supplementary Figs. 6, 7.
ᵇM*i* denotes the Mo atom at the M*i* site.
ᶜDFT-calculated peak position under 18.79 T was obtained using the correlation in Supplementary Fig. 5a, including the second-order quadrupolar shift.
ᵈQuadrupolar coupling constant |$C_Q$|. Experimental |$C_Q$| was estimated from the NMR spectrum (Fig. 3b) with DMFIT[69].

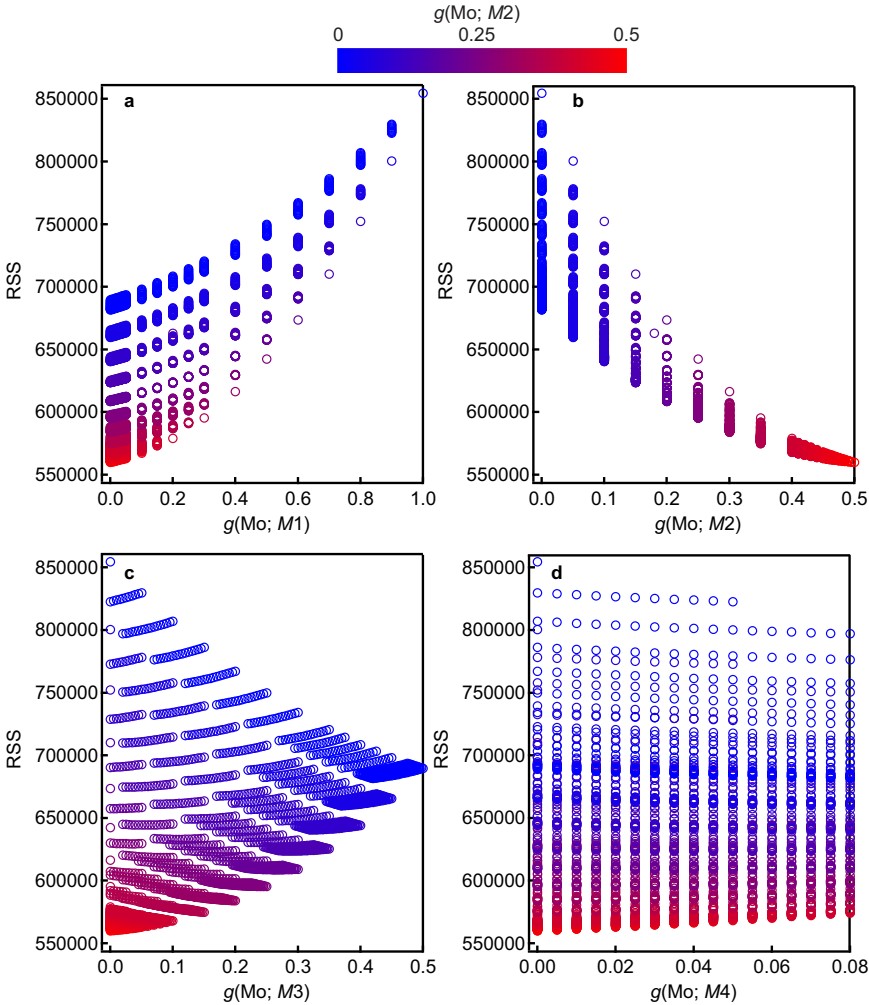

**Fig. 4 | Determination of the Mo occupancy factors in $Ba_7Nb_4MoO_{20} \cdot 0.15\ H_2O$.** Variation of the residual sum of squares (RSS; see the definition of Eq. (3)) with the occupancy factor of Mo atom at the **a** $M1$, **b** $M2$, **c** $M3$ and **d** $M4$ sites in the Rietveld analyses for the RXRD data measured with 0.6527887(5) Å X-ray at the BL02B2 beamline.

different sites were not observed in the experimental $^{95}$Mo NMR spectrum. Thus, the single $^{95}$Mo NMR peak was assigned to the $M2$ site. Similarly, observed $^{93}$Nb peaks at isotropic chemical shifts $\delta_{iso}$ = −748, −952 and −928 ppm can be assigned to the $M1$, $M2$ and $M3$ sites, respectively (Supplementary Table 4). These results lead us to conclude that the Mo cations are located at the $M2$ site near the ion-conducting c′ layer, indicating Mo order in $Ba_7Nb_4MoO_{20} \cdot 0.15\ H_2O$.

**Quantitative determination of the occupancy factors of Mo and Nb atoms by resonant X-ray diffraction**

We used resonant X-ray diffraction (RXRD) to quantify the occupancy factors of the Mo and Nb atoms in $Ba_7Nb_4MoO_{20} \cdot 0.15\ H_2O$. We measured the X-ray absorption near edge structure (XANES) spectra of $Ba_7Nb_4MoO_{20} \cdot 0.15\ H_2O$ and the resonant (anomalous) scattering factors of Nb atoms (Supplementary Table 5) were determined by Kramers−Kronig transformation from the XANES spectra[42] (Supplementary Fig. 8). In the Rietveld analyses of the RXRD data of $Ba_7Nb_4MoO_{20} \cdot 0.15\ H_2O$, we used the linear constraints Eq. (2), which were obtained in the preliminary analyses of the ND and conventional SXRD data. The occupancy factors of the Nb and Mo atoms at the $M1$, $M2$, $M3$ and $M4$ sites were not simultaneously refined because of strong correlations. Therefore, we carefully examined the residual sum of squares (RSS) in the Rietveld analysis for fixed occupancy values of Mo atoms at the $Mi$ site

$g(Mo; Mi)$ step-by-step (0.005 step interval for the finest case). Here the RSS is defined as

$$RSS = \sum_{i=1}^{N} \frac{1}{y_i^{obs}} \left( y_i^{obs} - y_i^{cal} \right)^2 \qquad (3)$$

where $N$, $y_i^{obs}$ and $y_i^{cal}$ are the total number of intensity data, the observed and calculated intensities for the $i^{th}$ step, respectively, of the RXRD data. Figure 4 shows RXRD results obtained with 0.6527887(5) Å X-ray at the BL02B2 beamline of SPring-8, which indicates that the occupancies of Mo atoms are 0.00 at the $M1$, $M3$ and $M4$ sites and 0.50 at the $M2$ site:

$$g(Mo; M1) = g(Mo; M3) = g(Mo; M4) = 0.00,\ g(Mo; M2) = 0.50 \qquad (4)$$

$$\text{Thus, } g(Nb; M1) = g(Nb; M3) = 1.00,$$
$$g(Nb; M2) = 0.42,\ g(Nb; M4) = 0.08 \qquad (4')$$

The same values were also obtained also in the Rietveld analyses for the RXRD data taken with a 0.6523630(5) Å X-ray at the different beamline BL19B2 of SPring-8 (Supplementary Fig. 9), validating the Mo occupancy values of Eq. (4). In preliminary analyses, the refined occupancy factors $g(Mo; Mi)$ ($i$ = 1, 3 and 4) were negative (Supplementary Tables 6–8), supporting the Mo occupancy factors of Eq. (4).

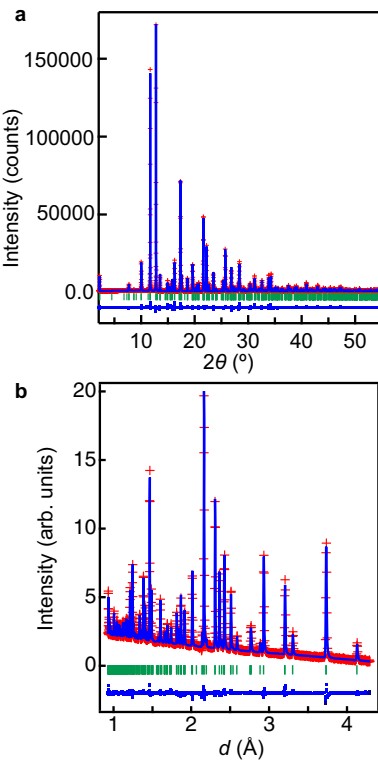

**Fig. 5 | Rietveld fitting patterns of $Ba_7Nb_4MoO_{20} \cdot 0.15\ H_2O$. a** Resonant X-ray diffraction (RXRD) data measured at 297 K with 0.6527887(5) Å X-ray at the BL02B2 beamline. **b** Neutron diffraction data at 300 K. The observed and calculated intensities and difference plots are shown by red cross marks, blue solid lines, and blue dots, respectively. Green tick marks denote the calculated Bragg peak positions.

These results clearly indicate the Mo chemical order at the $M2$ site near the ion-conducting c′ layer, which is consistent with the NMR results previously discussed.

### Complete crystal structure of $Ba_7Nb_4MoO_{20} \cdot 0.15\ H_2O$

To accurately refine the structural parameters of hydrogen and oxygen atoms, we have analysed the crystal structure of $Ba_7Nb_4MoO_{20} \cdot$ $0.15\ H_2O$ using neutron diffraction (ND) data collected at 30 and 300 K. During this process, the occupancy factors of Mo and Nb atoms were fixed to the values of Eqs. (4) and (4′), respectively, which were obtained from the analysis of the RXRD data. Excellent fittings were obtained for both ND and RXRD data (Fig. 5 and Supplementary Fig. 10). The crystallographic parameters refined using ND and RXRD data were consistent with each other (Table 2 and Supplementary Table 9). The water content $x$ in bulk crystalline $Ba_7Nb_4MoO_{20-x}(OH)_{2x}$ $(= Ba_7Nb_4MoO_{20+x}H_{2x} = Ba_7Nb_4MoO_{20} \cdot x\ H_2O)$ was calculated to be $x = 0.151(5)$ using the refined occupancy factors at 30 K (Supplementary Table 10), which is consistent with the water content estimated from the thermogravimetric-mass spectroscopic (TG-MS) analyses (Supplementary Fig. 11). The O1–H distance was estimated to be 1.07(4) Å using the refined crystal structure of $Ba_7Nb_4MoO_{20} \cdot x\ H_2O$ at 300 K, which agrees with the O–H distance of 0.99738(8) Å obtained from its Raman spectrum (Supplementary Fig. 12), indicating the presence of hydroxide ions formed by the hydration. The bond-valence sums (BVSs) of cations and anions for the refined structure of $Ba_7Nb_4MoO_{20} \cdot 0.15\ H_2O$ agree with their formal charges (Table 2). These results confirm the validity of the refined crystal structure of $Ba_7Nb_4MoO_{20} \cdot 0.15\ H_2O$.

Figure 6 shows the refined crystal structure of $Ba_7Nb_4MoO_{20} \cdot$ $0.15\ H_2O$, with the sequence c′hhcchh. Oxygen-deficient lattice O1 and

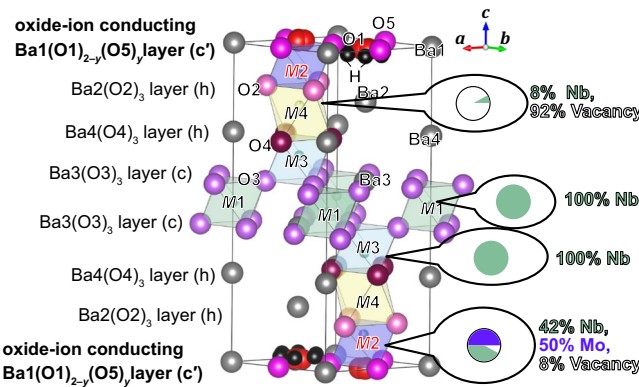

**Fig. 6 | Crystal structure of $Ba_7Nb_4MoO_{19.849}(OH)_{0.302}$.** Refined crystal structure of $Ba_7Nb_4MoO_{19.849}(OH)_{0.302}$ at 300 K, which shows the Mo chemical order and site occupancies of Nb and Mo atoms.

interstitial O5 sites exist in the c′ layer. At high temperatures, oxide ions can migrate via O1–O5 diffusion pathways and the interstitialcy diffusion mechanism as shown by the maximum-entropy method (MEM) neutron scattering length density (NSLD) distribution of $Ba_7Nb_{3.9}Mo_{1.1}O_{20.05}$ at 1073 K[36]. Similar O1–O5 paths were visualised in MEM NSLD distribution of wet $Ba_7Nb_4MoO_{20} \cdot 0.87\ H_2O$ at 368 K[34].

Structural disorders have been reported in $Ba_7Nb_4MoO_{20}$-based materials[11,34,36,38,39]. In contrast, a striking feature is the presence of Mo atoms only at the $M2$ site near the ion-conducting c′ layer, indicating Mo chemical order. DFT-optimised structures with Mo order at the $M2$ site have slightly lower energies than those with Mo disorder and Mo atoms at $M1$ and $M3$ sites, which supports the Mo chemical order at the $M2$ site (Supplementary Table 11). This is the first report on the chemical order of Mo atoms in $Ba_7Nb_4MoO_{20}$-based materials. In the literature[11,34–36,38,39], all structural analyses were performed based on complete Mo/Nb disorder. Meanwhile, in this study, the Mo order was indicated not only by structural refinements using RXRD data but also by NMR measurements and DFT calculations. An important question is why the Mo order occurs. The probable explanation is as follows: the $M2$ site has a smaller space compared with other $Mi$ sites ($i = 1, 3$ and 4) (Supplementary Table 12), and the size of the Mo cation is smaller than that of the Nb cation; thus, Mo order occurs. Indeed, the BVS of Mo at $M2$ site 5.54 agrees with the formal charge 6 of $Mo^{6+}$, which is higher than the BVS values of Mo atoms at $M1$ (4.63), at $M3$ (4.76) and at $M4$ (3.51) sites indicating the underbonding and instability of Mo atoms at $Mi$ sites ($i = 1, 3$ and 4).

## Discussion

The present work has demonstrated the chemical order of Mo atoms at the $M2$ site near the ion-conducting c′ layer in $Ba_7Nb_4MoO_{20} \cdot 0.15\ H_2O$ by the combined technique of solid-state NMR, resonant XRD and DFT calculations, in addition to the neutron diffraction and conventional SXRD. The NMR spectra provided direct experimental evidence for the Mo order, while the structural analyses using the RXRD data enabled the quantitative determination of the occupancy factors of Mo and Nb atoms. This combined technique can be used to investigate the hidden chemical order in various ion-conducting hexagonal perovskite derivatives such as $Ba_7Nb_{4-x}Mo_{1+x}O_{20+x/2}$[36], $Ba_7Nb_{4-x}W_xMoO_{20+x/2}$[38], $Ba_7Nb_{4-x}Cr_xMoO_{20+x/2}$[39] and $Ba_3MoNbO_{8.5}$[6,43,44] where the Mo occupancies at the $Mi$ sites ($i = 1, 2, 3$ and 4) are unknown. Here $x$ is the dopant or excess Mo content. Beyond the limits of the combined technique of conventional X-ray diffraction and NMR ('SMARTER' crystallography[45,46]), this RXRD/NMR method can be applied to numerous compounds such as thermoelectric $Ag_{1-x}Cd_xSbTe_2$[16] and superconducting $Zr_5Ir_2Os$[47] exhibiting chemical order/disorder of atoms with both similar X-ray atomic scattering factors and similar

**Table 2 | Refined crystal parameters and reliability factors in Rietveld analysis of the neutron diffraction data of Ba$_7$Nb$_4$MoO$_{19.849}$(OH)$_{0.302}$ (= Ba$_7$Nb$_4$MoH$_{0.302}$O$_{20.151}$ = Ba$_7$Nb$_4$MoO$_{20.151}$H$_{0.302}$ = Ba$_7$Nb$_4$MoO$_{20}$·0.151 H$_2$O) at 300 K**

| Site / Atom label | Atom | Wyckoff position | $g^f$ | $x$ | $y$ | $z$ | $U_{iso}$ (Å$^2$)$^g$ | BVS$^d$ |
|---|---|---|---|---|---|---|---|---|
| Ba1 | Ba | 1a | 1$^e$ | 0 | 0 | 0 | 0.0165(7) | 2.02 |
| Ba2 | Ba | 2d | 1$^e$ | 1/3 | 2/3 | 0.82374(7) | 0.0138(3) | 2.19 |
| Ba3 | Ba | 2d | 1$^e$ | 1/3 | 2/3 | 0.57420(9) | 0.0091(3) | 2.29 |
| Ba4 | Ba | 2c | 1$^e$ | 0 | 0 | 0.27870(8) | 0.0078(4) | 1.91 |
| $M1$ | Nb | 1b | 1$^e$ | 0 | 0 | 1/2 | 0.0060(3) | 4.57 |
| $M2$ | Nb | 2d | 0.42 | 1/3 | 2/3 | 0.09489(6) | 0.0060(3) | 4.91 |
| $M2$ | Mo | 2d | 0.5 | 1/3 | 2/3 | 0.09489(6) | 0.0060(3) | 5.54 |
| $M3$ | Nb | 2d | 1$^e$ | 1/3 | 2/3 | 0.34909(6) | 0.0060(3) | 4.61 |
| $M4$ | Nb | 2d | 0.08 | 1/3 | 2/3 | 0.1926$^a$ | 0.0060(3) | 3.65 |
| O1 | O | 6i | 1/3 | 0.3532(4) | 0.7064(9) | –0.01209(6) | 0.0196(8) | 1.84 |
| O2 | O | 6i | 1$^e$ | 0.16652(14) | 0.3330(2) | 0.13082(4) | 0.01167(16) | 1.94 |
| O3 | O | 6i | 1$^e$ | 0.16323(14) | 0.3265(2) | 0.43098(4) | 0.01086(18) | 1.95 |
| O4 | O | 6i | 1$^e$ | 0.49502(9) | 0.50498(9) | 0.29455(3) | 0.00791(18) | 1.98 |
| O5 | O | 3e | 0.0504$^b$ | 1/2 | 0 | 0 | 0.0196(8) | 1.28 |
| H | H | 12j | 0.0252$^b$ | 0.346(4) | 0.500(5) | 0.9748(16) | 0.041$^c$ | 0.85 |

Crystal system: trigonal. Space group: $P\bar{3}m1$ (No.164, setting 1). Lattice parameters: $a = b = 5.865653(4)$ Å, $c = 16.53699(3)$ Å. The number of formula per unit cell: $Z = 1$. Reliability factors: $R_{wp} = 2.719\%$, $R_p = 2.254\%$, $R_B = 4.577\%$, $R_F = 3.872\%$, GoF = 19.857.

$^a$z coordinate of the Nb4 atom was fixed to those from preliminary analyses.

$^b$Occupancy factors of O5 and H atoms were fixed to those from ND analysis at 30 K.

$^c$Atomic displacement parameter of the H atom was fixed to those from preliminary analyses.

$^d$BVS bond-valence sums. Here the bond-valence parameters after ref. 70 were used for the calculations of BVSs. The low BVS values of O5 and H atoms than the formal charges of –2 and +1 are consistent with the low occupancy values of 0.0504 and 0.0252, respectively.

$^e$The occupancy factors of Ba1–Ba4, $M1$, $M3$, and O2–O4 atoms were fixed to unity, because the refined values agreed with unity within three times of estimated standard deviations (see Supplementary Note 1 for details).

$^f$$g = g(X; s)$: Occupancy factor of $X$ atom at the $s$ site. $g$(Ba; Ba1) = $g$(Ba; Ba2) = $g$(Ba; Ba3) = $g$(Ba; Ba4) = $g$(Nb; $M1$) = $g$(Nb; $M3$) = $g$(O; O2) = $g$(O; O3) = $g$(O; O4) = 1; $g$(Nb; $M2$) = 0.42, $g$(Mo; $M2$) = 0.5, $g$(Nb; $M4$) = 0.08; $g$(O; O1) = 1/3, $g$(O; O5) = 0.0504; $g$(H; H) = 0.0252. $x$, $y$, and $z$: atomic coordinates.

$^g$$U_{iso}$($Xn$) Isotropic atomic displacement parameter of $X$ atom at the $Xn$ site. Linear constraints in the Rietveld analysis: $U_{iso}$(Nb1) = $U_{iso}$(Nb2) = $U_{iso}$(Mo2) = $U_{iso}$(Nb3) = $U_{iso}$(Nb4).

neutron scattering lengths (Fig. 1 and Supplementary Table 1). In contrast to the single-crystal X-ray diffraction[19,48] and X-ray fluorescence holography[49], the RXRD/NMR method uses powders or polycrystalline samples, making it versatile and easily applicable to in situ measurements (e.g., at high temperatures). The combined technique would be useful for investigating not only the periodic average structure but also the short- and intermediate-range order/disorder hidden in conventional diffraction and total scattering.

Next, we discuss the influences of Mo chemical order on the material properties of Ba$_7$Nb$_4$MoO$_{20}$. The flexibility of the coordination of $M2$ atoms near the c′ layer was suggested to determine the high ion conduction in Ba$_7$Nb$_4$MoO$_{20}$·0.5 H$_2$O from the ab initio molecular dynamics simulations[34]. Since the present work has indicated that Mo cations are localised at the $M2$ site near the ion-conducting c′ layer in Ba$_7$Nb$_4$MoO$_{20}$·0.15 H$_2$O, the flexibility of Mo atoms is important for the high ion conduction in Ba$_7$Nb$_4$MoO$_{20}$-based materials as well as in other Mo-containing ionic conductors such as La$_2$Mo$_2$O$_9$[50]. Therefore, the bulk conductivity of Ba$_7$Nb$_{4-x}$Mo$_{1+x}$O$_{20+x/2}$ increases with increasing the excess amount of Mo atoms $x$ from $x = 0$ to 0.1[36], which is ascribed to not only a larger amount of excess oxygen atoms but also the larger amount of Mo atoms.

The energy barriers for oxide-ion migration $E_{b/O}$ of Mo-ordered and virtual Mo-disordered Ba$_7$Nb$_4$MoO$_{20}$·0.15 H$_2$O were calculated using the bond-valence method[51,52]. The $E_{b/O}$ along the c axis in Mo-ordered Ba$_7$Nb$_4$MoO$_{20}$·0.15 H$_2$O (1.93 eV) is higher than that in the virtual Mo-disordered Ba$_7$Nb$_4$MoO$_{20}$·0.15 H$_2$O (1.60 eV) [Supplementary Table 13], which is attributable to the narrower bottleneck for oxide-ion migration along the c axis due to the higher occupancy factor of larger Nb cations at the bottleneck triangle (Supplementary Fig. 14).

The substitution of Nb with Mo improves the oxide-ion conductivity because of the larger number of interstitial oxygen atoms (higher carrier concentration). The formation energies $\Delta H_f$ of the Mo-ordered and virtual Mo-disordered Ba$_7$Nb$_{3.5}$Mo$_{1.5}$O$_{20.25}$ oxides were calculated using the DFT method. The calculated $\Delta H_f$ values of Mo-ordered models are lower than those of virtual Mo-disordered ones (Supplementary Table 15), which indicates that Mo ordering stabilises Ba$_7$Nb$_{3.5}$Mo$_{1.5}$O$_{20.25}$ with interstitial oxygen atoms more efficiently than Mo disordering, leading to higher oxide-ion conductivity. The hydration enthalpies $\Delta H_{hyd}$ of Mo-ordered and Mo-disordered Ba$_7$Nb$_4$MoO$_{20}$ were also investigated by DFT calculations, because the hydration is important for proton conduction in Ba$_7$Nb$_4$MoO$_{20}$. Compared with the calculated $\Delta H_{hyd}$ of the Mo-disordered Ba$_7$Nb$_4$MoO$_{20}$ (1.70 kJ mol$^{-1}$), that of Mo-ordered Ba$_7$Nb$_4$MoO$_{20}$ (−22.7 kJ mol$^{-1}$) is close to the experimental value below 300 °C (−24 kJ mol$^{-1}$)[34]. The calculated $\Delta H_{hyd}$ for the Mo-ordered system (−22.7 kJ mol$^{-1}$) is lower than that of the Mo-disordered one (1.70 kJ mol$^{-1}$), indicating that the Mo ordering also stabilises the hydrated Ba$_7$Nb$_4$MoO$_{20}$ more efficiently compared with Mo disordering. These results demonstrate that the Mo order in Ba$_7$Nb$_4$MoO$_{20}$ affects the material properties. The present findings represent a major advance in the fundamental understanding of the correlation between the crystal structure and material properties of ionic conductors.

## Methods

### Synthesis and characterisation

The Ba$_7$Nb$_4$MoO$_{20}$·0.15 H$_2$O samples were prepared by the solid-state reaction method. High-purity (>99.9%) BaCO$_3$, Nb$_2$O$_5$ and MoO$_3$ were mixed as ethanol slurries and ground as dry powders using an agate mortar and pestle. The obtained powders were calcined at 900 °C for 12 h for decarbonation. The materials thus obtained were crushed and ground into fine powders in an agate mortar for 1 h as dried powders and ethanol slurries. The resultant powders were uniaxially pressed at 150 MPa and then sintered in air at 1100 °C for 24 h. The sintered pellets were crushed and ground into fine powders for X-ray powder

diffraction (XRD), inductively coupled plasma atomic emission spectroscopy (ICP-AES, Shimadzu ICPS-8100 spectrometer), and TG-MS measurements. The ICP-AES results indicated that the cation molar ratio of $Ba_7Nb_4MoO_{20} \cdot 0.15\,H_2O$ was Ba: Nb: Mo = 6.89(12): 4.078(18): 1.034(10), which is consistent with the nominal composition. TG-MS analyses of $Ba_7Nb_4MoO_{20} \cdot 0.15\,H_2O$ were performed using RIGAKU Thermo Mass Photo under He flows at a heating rate of 20 K min$^{-1}$ up to 900 °C. The Raman spectrum of $Ba_7Nb_4MoO_{20} \cdot 0.15\,H_2O$ was collected using an NRS-4100 (JASCO Co.) instrument with an excitation wavelength of 532 nm.

### Synchrotron X-ray and neutron diffraction experiments and data analysis

Synchrotron X-ray diffraction (SXRD) experiments were performed at beamlines BL02B2 (297 K)[53] and BL19B2 (300 K)[54] of SPring-8. X-ray wavelengths for resonant X-ray diffraction experiments were selected from the spectrum of Nb K-edge X-ray absorption near edge structure (XANES) for a $Ba_7Nb_4MoO_{20} \cdot 0.15\,H_2O$ powder. X-ray wavelengths were determined from the X-ray diffraction data of standard silicon powder (SRM 640c) using FullProf software[55]. Conventional SXRD data were recorded with a 0.6994806(5) Å X-ray. RXRD measurements were performed using a 0.6527887(5) Å X-ray at the BL02B2 beamline and a 0.6523630(5) Å X-ray at BL19B2. Both the conventional SXRD and RXRD data were analysed using the Rietveld method with the computer programme RIETAN-FP[56]. We used atomic scattering factors in the form of $f = f_0 + f' + if''$, where $f_0$ is the Thomson scattering factor and $f'$ and $f''$ are the resonant (anomalous) scattering factors. The $f'$ and $f''$ factors of the Nb atom were calculated from the XANES spectrum (Supplementary Fig. 8) recorded at BL19B2 with the programme DiffKK[57], and the $f'$ and $f''$ factors of Ba, Mo and O atoms were obtained from the theoretical values reported by Cromer and Libermann[58] (Supplementary Table 5).

Time-of-flight (TOF) neutron diffraction data of $Ba_7Nb_4MoO_{20} \cdot 0.15\,H_2O$ were obtained at 30 and 300 K using a high-intensity total diffractometer NOVA (BL-21) in the MLF of J-PARC. Rietveld analyses were performed using Z-Rietveld[59,60] using neutron diffraction data obtained from the backscattering bank of the NOVA.

The bond-valence-based energy (BVE) landscapes for a test oxide ion and proton in $Ba_7Nb_4MoO_{20} \cdot 0.15\,H_2O$ were calculated using refined crystal parameters at 300 K using the SoftBV programme[51,52]. The refined structures and BVE landscapes were depicted using the VESTA 3[61].

### Solid-state NMR experiments

NMR experiments of $Ba_7Nb_4MoO_{20} \cdot 0.15\,H_2O$ were performed with a 3.2-mm homemade MAS probe at a spinning speed of 20 kHz under a magnetic field of 18.79 T, corresponding to $^{95}$Mo and $^{93}$Nb Larmor frequencies of 52.16 and 195.84 MHz, respectively. 1D $^{95}$Mo and 2D $^{93}$Nb NMR spectra were recorded using a JEOL JNM-ECA 800 spectrometer, whereas 1D $^{93}$Nb NMR spectra were obtained using a JEOL JNM-ECZ 800 R spectrometer. $^{95}$Mo chemical shifts were referenced to 2.0 M aqueous solution of $Na_2MoO_4$ at 0.00 ppm (refs. 62,63), and $^{93}$Nb chemical shifts were externally referenced to $NaNbO_3$ at −1093 ppm (ref. 64). $^{95}$Mo NMR spectra of α-$MoO_3$ and $BaMoO_4$ were also obtained to investigate the relationships between the experimental and DFT-calculated NMR parameters (Supplementary Figs. 4, 5). The 1D $^{95}$Mo NMR spectra were acquired by accumulating 22,000 scans using a 1.2 μs single-pulse sequence with a relaxation delay of 20 s. The 1D $^{93}$Nb spectra were measured using a spin-echo sequence (2.0 and 4.0 μs), accumulating 1024 scans with a relaxation delay of 1 s. The 2D $^{93}$Nb 3QMAS NMR spectra were measured by the conventional three-pulse sequence with z-filter[65] (2.0, 0.9 and 15 μs), and recorded with 264 transients averaged for each of the 1024 $t1$ points with a relaxation delay of 0.2 s. Shearing transformation[66] was applied to the spectra. Here, the centre of the F1 axis was set to the centre of the F2 axis.

### Density functional theory (DFT)-based calculations

Generalised gradient approximation (GGA) electronic calculations were performed using Vienna Ab initio Simulation Package (VASP)[41]. We used projector augmented wave (PAW) potentials for Ba, Nb, Mo and O atoms, plane-wave basis sets with a cutoff of 500 eV and the Perdew–Burke–Ernzerhof (PBE) GGA functionals. The crystal parameters refined using the neutron diffraction data of $Ba_7Nb_4MoO_{20} \cdot 0.15\,H_2O$ at 300 K were used as the initial parameters in the DFT structural optimisations. Atomic coordinates of $Ba_7Nb_4MoO_{20}$ were optimised in the space group $P1$, with the convergence condition of 0.02 eV Å$^{-1}$. The supercell programme[67] was used to generate supercell models. The formation energies of $Ba_7Nb_{3.5}Mo_{1.5}O_{20.25}$ $\Delta H_f$ for the Mo-ordered and virtual Mo-disordered models were calculated according to the following equation:

$$Ba_7Nb_4MoO_{20} + 1/2MoO_3 \rightarrow Ba_7Nb_{3.5}Mo_{1.5}O_{20.25} + 1/4Nb_2O_5$$

The optimised structures are shown in Supplementary Fig. 15. The hydration enthalpies $\Delta H_{hyd}$ were also estimated for the Mo-ordered and virtual Mo-disordered models (Supplementary Fig. 16) according to the following reaction:

$$(Ba_7Nb_4MoO_{20})_4 + H_2O \rightarrow (Ba_7Nb_4MoO_{20.25}H_{0.5})_4$$

DFT calculations of the $^{93}$Nb and $^{95}$Mo chemical magnetic shielding and electric field gradient tensors were performed using the VASP code with a cutoff energy of 700 eV for the plane-wave basis sets, where the total energy converged within $10^{-8}$ eV/atom. The GIPAW formalism[28] was utilised for the calculations of the NMR chemical shielding tensors.

## Data availability

The datasets generated during and/or analysed during the current study are available from the corresponding author on request.

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

## Acknowledgements

We are grateful to Mr. K. Deguchi and Dr. T. Shimizu of the National Institute for Materials Science for their assistance with NMR measurements. We thank Dr. K. Chiba, Dr. W. Zhang, Dr. T. Murakami, Dr. H. Yaguchi, Mr. K. Saito, Ms. R. Morikawa, Mr. Y. Suzuki, Mr. K. Jojima, Mr. M. Miyazawa and Mr. K. Matsuzaki for useful discussion and assistance in the experiments/analyses. We also thank Ms. K. Suda and Dr. M. Tada of the Materials Analysis Division, Open Facility Center, Tokyo Institute of Technology, for their assistance in the TG-MS and Raman measurements, respectively. We acknowledge Kojundo Chemical Laboratory Co. for the ICP measurements and for supplying chemical materials. Neutron and synchrotron X-ray experiments were performed under project Nos. 2019BF2106, 2020L0801, 2020L0804, 2019A1052, 2020A1730, 2021A1599 and 2021B1826. This study and travel expenses were supported by a Grant-in-Aid for Scientific Research (KAKENHI, Nos. JP19H00821 (M.Y.), JP20J23124 (Y.Y.), JP21J22818 (Y.S.), JP21K18182 (M.Y.) and JP22H04504 (K.F.)) from the Ministry of Education, Culture, Sports, Science and Technology of Japan, Adaptable and Seamless Technology Transfer Programme through Target-driven R&D (A-STEP) from the Japan Science and Technology Agency (JST) Grant Number JPMJTR22TC (M.Y.), and JSPS Core-to-Core Programme, A. Advanced Research Networks ([i] Solid Oxide Interfaces for Faster Ion Transport (M.Y.) and [ii] Mixed Anion Research for Energy Conversion [JPJSCCA20200004] (M.Y.)). Y.Y. and Y.S. acknowledge support in the form of a JSPS Fellowship for Young Scientists DC1 (20J23124 and 21J22818). A part of this work was supported by NIMS microstructural characterisation platform as a programme of the 'Nanotechnology Platform' of the Ministry of Education, Culture, Sports, Science and Technology (MEXT), Japan, Grant Numbers JPMXP09A19NM0110 (M.Y.). This work contains the result of using research equipment shared in the MEXT Project for promoting public utilisation of advanced research infrastructure (Programme for supporting the introduction of the new sharing system) Grant Number JPMXS0420900521.

## Author contributions

Y.Y. and M.Y. designed research. Y.Y. and Y.S. prepared the samples and measured TG data. M.T., A.G., S.O. and Y.M. measured NMR data. Y.Y., K.F., Y.S., S.KO., S.KA. and K.O. measured synchrotron XRD data. Y.Y. and T.I. performed the DFT calculations. Y.Y., K.F., Y.S., K.I. and T.O. collected the neutron diffraction data. Y.Y. analysed the crystal structure and Raman data. Y.Y., M.T., K.F. and M.Y. wrote the original draft of the manuscript. M.Y. and Y.Y. edited the manuscript. M.Y. conceived the project and supervised the research. All authors participated in the data analysis, discussed the results and read the manuscript.

## Competing interests

The authors declare no competing interests.
