## [Peer Review File · Nature Communications]

Hidden chemical order in disordered Ba₇Nb₄MoO₂₀ revealed by resonant X-ray diffraction and solid-state NMRREVIEWER COMMENTS

Reviewer #1 (Remarks to the Author):

This manuscript combines resonant X-ray diffraction with solid-state NMR experiments and DFT calculations to understand chemical order in a solid barium niobium molybdenum oxide. The work convincingly shows that molybdenum sits in an ordered fully occupied site while the niobium cations occupy three other sites. The authors make the case for understanding the properties of the material in the context of this order/disorder. I have the following questions/concerns:

1. The English and wording in some places needs improvement prior to publication.
2. For much of the manuscript, the authors refer to an anhydrous compound, but then introduce the fact that it is a hydrate. The correct formula should be used throughout. Do different levels of hydration influence the results found here?
3. line 89: make it clear why you are concluding that Mo order facilitates high conductivities.
4. There is a typo in the graphical abstract: chemiral.
5. Show your fitting of the 95 Mo NMR spectrum. Can the satellites be observed to improve precision? Models other than the M2 site give DFT-computed quadrupolar coupling constants close to 1 to 2 MHz, so a good fit of the experimental data is needed to confirm that the M2 site is the right one.
6. Figure S5 part a and part d: please clarify shift vs shielding. I believe you are plotting shielding on the x axes but this is labelled as 'shift'.

Reviewer #2 (Remarks to the Author):

This paper reports on hidden chemical order in the oxide ion conductor $\text{Ba}_7\text{Nb}_4\text{MoO}_{20}$. The authors have used a range of techniques to determine this as it is not possible via conventional neutron and/or X-ray diffraction. This new method could potentially be applied to other systems. I am not an expert in NMR but it appears to show clear Mo order with Mo on a single site. DFT calculations back up the NMR results nicely. It is very interesting that the Mo is only found in the polyhedra where the ionic conductivity arises and it is suggested this arises due to the smaller size of Mo^{6+} compared to Mo^{5+} which is plausible.

The results are interesting and could potentially be published in Nature Communications but more insight is needed into the significance to the field and related fields.

Further comments are:

- 1) The English needs checking throughout. Some of the sentences in the abstract need rewording – it's a little jumbled. There is some repetition in parts also.
- 2) The energy barriers for oxide ion and proton conduction are higher in the "ordered" Mo system compare to the "virtual disordered". How do you reconcile this with your results? More thought is needed here.
- 2) How do these results offer unprecedented insight into the understanding of the ion diffusion mechanism in hexagonal perovskite-related oxides. Can you predict anything from this? Would there be an optimum Nb:Mo ratio for obtaining high ionic conductivity? What other systems could this be used for and why is it so important? How is this method different/better than other local structure studies?

Reviewer #3 (Remarks to the Author):

The paper by Masatomo Yashima and co-authors is an interesting work on the detailed structural analysis of the high ion-conducting $\text{Ba}_7\text{Nb}_4\text{MoO}_{20}$, which unravels the crystallographic order/disorder of Mo and Nb in the crystal lattice. The authors used a judicious combination of advanced techniques, including high-field solid-state NMR, resonant X-ray and neutron diffractions, and DFT calculations, to determine the precise location of Mo and Nb among the four possible crystallographic sites and their occupancy rates. Although this methodology is known as SMARTER

crystallography (Structure elucidation by coMbinIng mAgnetic Resonance, compuTation modELing and diffRactions), (see e.g. DOI : 10.1039/C2DT30100H), it should be further developed to solve challenging structural problems such as the one presented in this manuscript. Here, the main experimental evidence comes from ^{95}Mo NMR and resonant X-ray diffraction, supported by DFT calculations of a number of structural models. The article is very well written. I find this work very interesting and recommend that it be accepted for publication in Nature Communications.

However, I have a few questions to clarify:

1- Lines 2, 33, 58, and 179: I think "resonance" X-ray diffraction is not appropriate term. It should be replaced by "resonant". Also in Fig. captions !!

2- Line 83: The cell parameters are not complete (missing "b") and do not match those in Supplementary Figures 6 & 7. This is also in Table 2, and in Supplementary Tables 9 & 10. Please check.

3- Fig. 1 caption: It is said, "All the SCS values are available in the Supplementary data, but we can see only some of them in the Supplementary Table 1.

4- Table 2: "a" and "b" are present in the endnote but missing from the Table. The same for Supplementary Table 9.

5- Supplementary Fig. 3: The assignment of the ^{93}Nb NMR is questionable, even though is based on DFT calculations. The problem is that the observed chemical shifts do not follow the expected trend with coordination number (CN), (see doi:10.1016/j.ssnmr.2005.09.003). At first glance, I would say that the M2 site with CN 4 should correspond to the signal at -748 ppm and those at -952 and -928 ppm with CN 6 both, to M1 and M2 sites. Is it possible to integrate or calculate the area of these signals? M2 should account for 1 Nb and M2 + M3 for 4 Nb. This is likely to be difficult with the current spectrum shown in Supplementary Fig. 3 because of the overlapping sidebands with the signals. A higher magnetic field and faster MAS will improve resolution and solve this problem. Moreover, the presence of an oxide impurity is surprising for this supposedly very pure compound. An elemental analysis is strongly recommended to settle this point.

6- Supplementary Tables 2 & 3: In the "c" endnote, it should be "supplementary Fig. 5" (and not 6).

RESPONSE TO REVIEWERS' COMMENTS

We are submitting the revised version of our manuscript. We appreciate the reviewers and editors for their careful reading and helpful suggestions. We have considered their feedback and incorporated their suggestions into the revised manuscript. All the changes for the action for the suggestions are **highlighted by yellow** and other changes (mainly by the English editing service editage) are written **in red** in the revised manuscript (MS_withTracks.docx) and revised SI (SI_withTracks.docx), which are files only for review.

We hope that our response and the revised manuscript satisfactorily address the reviewers' comments and suggestions.

Masatomo Yashima (Tokyo Institute of Technology)
on behalf of all authors

See the next pages.

Response to the Reviewer #1.

Reviewer's comment: This manuscript combines resonant X-ray diffraction with solid-state NMR experiments and DFT calculations to understand chemical order in a solid barium niobium molybdenum oxide. The work convincingly shows that molybdenum sits in an ordered fully occupied site while the niobium cations occupy three other sites. The authors make the case for understanding the properties of the material in the context of this order/disorder. I have the following questions/concerns:

1. The English and wording in some places needs improvement prior to publication.

Response: Thank you for the helpful comment.

Action: We used the professional English editing service Editage to correct grammatical errors and improve the wording, where the changed parts are shown **in the red**.

editage

Editing Certificate

This document certifies that the manuscript listed below has been edited to ensure language and grammar accuracy and is error free in these aspects. The edit was performed by professional editors at Editage, a division of Cactus Communications. The author's core research ideas were not altered in any way during the editing process. The quality of the edit has been guaranteed, with the assumption that our suggested changes have been accepted and the text has not been further altered without the knowledge of our editors.

MANUSCRIPT TITLE

Hidden chemical order in disordered Ba₇Nb₄MoO₂₀ revealed by resonant X-ray diffraction and solid-state NMR

AUTHORS

Yuta Yasui Masataka Tansho, Kotaro Fujii, Yuichi Sakuda, Atsushi Goto, Shinobu Ohki, Yuuki Mogami, Takahiro Iijima, Shintaro Kobayashi, Shogo Kawaguchi, Keiichi Osaka, Kazutaka Ikeda, Toshiya Otomo, Masatomo Yashima

ISSUED ON

February 06, 2023

JOB CODE

MLCQD_3

Vikas Narang

Vikas Narang
Chief Operating Officer - Editage

editage

Editage, a brand of Cactus Communications, offers professional English language editing and publication support services to authors engaged in over 1300 areas of research. Through its community of experienced editors, which includes doctors, engineers, published scientists, and researchers with peer review experience, Editage has successfully helped authors get published in internationally reputed journals. Authors who work with Editage are guaranteed excellent language quality and timely delivery.

GLOBAL :
+1(833) 979-0061 | request@editage.com

JAPAN :
0120-50-2987 | submissions@editage.com

CACTUS

 **impact.science**

 **researcher.life**

 **lifesciences.cactusglobal.com**

Reviewer's comment: 2. For much of the manuscript, the authors refer to an anhydrous compound, but then introduce the fact that it is a hydrate. The correct formula should be used throughout. Do different levels of hydration influence the results found here?

Response: Thank you for the useful comments. In this work, we have investigated only as prepared sample where the hydration occurs to some extent. Thus, we changed the chemical formula from $Ba_7Nb_4MoO_{20}$ to $Ba_7Nb_4MoO_{20} \cdot 0.15 H_2O$.

Dry samples were prepared by annealing the as-prepared samples at 900 °C for 12 h in dry air and put into a sealed glass capillary. RXRD experiments were also performed on the dry samples and the occupancy factors of Mo atoms in the dry $Ba_7Nb_4MoO_{20}$ was investigated by the Rietveld analysis of the RXRD data.

Figure A1. Variation of the residual sum of squares RSS with the occupancy factor of Mo atom at the M2 site in Rietveld analysis for RXRD data of the dry $Ba_7Nb_4MoO_{20}$ sample.

Table A1. Comparison of the occupancy factors $g(Mo; M1)$, $g(Mo; M2)$, $g(Mo; M3)$ and $g(Mo; M4)$ between the hydrated $Ba_7Nb_4MoO_{20} \cdot 0.15 H_2O$ and dry $Ba_7Nb_4MoO_{20}$ samples. These occupancy values have minimum RSS values in the plots of RSS against the occupancy factors (An example can be seen in Figure A1).

	Hydrated sample (data in the manuscript)	Dry sample
$g(Mo; M1)$	0.00	0.00
$g(Nb; M1)$	1.00	1.00
$g(Mo; M2)$	0.50	0.50
$g(Nb; M2)$	0.42	0.46
$g(Mo; M3)$	0.00	0.00
$g(Nb; M3)$	1.00	1.00
$g(Mo; M4)$	0.00	0.00
$g(Nb; M4)$	0.08	0.04

In both hydrated and dry samples, $g(\text{Mo}; M2) = 0.50$, $g(\text{Mo}; M1) = g(\text{Mo}; M3) = g(\text{Mo}; M4) = 0.00$, indicating that Mo is localized only at the $M2$ site in both samples. Therefore, the conclusion that Mo is ordered in the $M2$ site is valid in both hydrated and dry samples.

Action: The composition in the revised manuscript has been changed from “Ba₇Nb₄MoO₂₀” to “Ba₇Nb₄MoO₂₀·0.15 H₂O” where the changed parts are shown by yellow hatch.

- In this study, we aim to elucidate the Mo/Nb order/disorder in a high ion conductor Ba₇Nb₄MoO₂₀·0.15 H₂O using the combined technique of RXRD and NMR (Fig. 2a). We chose Ba₇Nb₄MoO₂₀·0.15 H₂O, because Ba₇Nb₄MoO₂₀-based oxides and related compounds are emerging materials with high ion conduction, structural disorder and high chemical stability^{5,34–40}.
- A single hexagonal phase of Ba₇Nb₄MoO₂₀·0.15 H₂O was prepared by the solid-state reactions (Supplementary Fig. 1). The lattice parameters of Ba₇Nb₄MoO₂₀·0.15 H₂O were determined to be $a = 5.8654(3)$ and $c = 16.5390(3)$ Å using the X-ray diffraction data of the mixture of Ba₇Nb₄MoO₂₀·0.15 H₂O sample and internal standard silicon. To determine the occupancy factors of Nb_{0.8}Mo_{0.2}, Ba and O atoms, preliminary Rietveld analyses of Ba₇Nb₄MoO₂₀·0.15 H₂O were performed using neutron diffraction (ND) data and conventional synchrotron X-ray diffraction (SXR) data recorded with 0.6994806(5) Å X-ray far from the Nb K edge, based on the Mo/Nb disordered model (Supplementary Note 1 and Supplementary Figure 2 for details). Ba₇Nb₄MoO₂₀·0.15 H₂O was confirmed to be a $P\bar{3}m1$ hexagonal perovskite polytype 7H with four Mo/Nb cation sites ($M1$, $M2$, $M3$, and $M4$) (Fig. 2b).
- The refined crystal parameters of Ba₇Nb₄MoO₂₀·0.15 H₂O were consistent with those reported in the literature^{11,38}.
- We performed ⁹³Nb and ⁹⁵Mo solid-state NMR experiments on Ba₇Nb₄MoO₂₀·0.15 H₂O at a high magnetic field (18.8 T), which enabled the selective observation of Nb and Mo cations, respectively²³.
- In contrast, in the 1D ⁹⁵Mo MAS NMR spectrum only one peak is observed (Fig. 3b), indicating a single Mo site and Mo order in Ba₇Nb₄MoO₂₀·0.15 H₂O.
- Thus, we can assign the Nb and Mo peaks by comparing the experimental and calculated NMR parameters of Ba₇Nb₄MoO₂₀·0.15 H₂O.
- The calculated peak positions for (Mo₂)O₄ tetrahedron of Ba₇Nb₄MoO₂₀ ranged from –29 to –36 ppm, which is close to the experimental peak position of –47 ppm for Ba₇Nb₄MoO₂₀·0.15 H₂O.
- These results lead us to conclude that the Mo cations are located at the $M2$ site near the ion-conducting c' layer, indicating Mo order in Ba₇Nb₄MoO₂₀·0.15 H₂O.
- We used resonant X-ray diffraction (RXRD) to quantify the occupancy factors of the Mo and Nb atoms in Ba₇Nb₄MoO₂₀·0.15 H₂O.
- In the Rietveld analyses using the RXRD data of Ba₇Nb₄MoO₂₀·0.15 H₂O, we used the linear constraints Eqs. (2), which were obtained in from preliminary analyses of the ND and conventional SXR data.
- The energy barriers for oxide-ion migration $E_{b/O}$ of Mo-ordered and virtual Mo-disordered Ba₇Nb₄MoO₂₀·0.15 H₂O were calculated using the bond-valence method^{50,51}.

The $E_{b/O}$ along the c axis in Mo-ordered $\text{Ba}_7\text{Nb}_4\text{MoO}_{20}\cdot 0.15 \text{H}_2\text{O}$ 1.93 eV is higher than that in the virtual Mo-disordered $\text{Ba}_7\text{Nb}_4\text{MoO}_{20}\cdot 0.15 \text{H}_2\text{O}$ 1.60 eV (Supplementary Table 13), which is attributable to the narrower bottleneck for oxide-ion migration along the c axis due to the higher occupancy factor of larger Nb cations at the bottleneck triangle (Supplementary Fig. 14).

- The $\text{Ba}_7\text{Nb}_4\text{MoO}_{20}\cdot 0.15 \text{H}_2\text{O}$ samples were prepared by the solid-state reaction method. High-purity (> 99.9%) BaCO_3 , Nb_2O_5 , and MoO_3 were mixed using an agate mortar and pestle as the ethanol slurry and dry powder, respectively.
- **Fig. 2. Strategies for the elucidation of the Mo/Nb order and complete crystal structure of $\text{Ba}_7\text{Nb}_4\text{MoO}_{20}\cdot 0.15 \text{H}_2\text{O}$.**
- **Fig. 3 Solid-state NMR spectra of $\text{Ba}_7\text{Nb}_4\text{MoO}_{20}\cdot 0.15 \text{H}_2\text{O}$, showing the site assignment and Mo order.**
- **Table 1. Calculated and experimental ^{95}Mo NMR parameters of $\text{Ba}_7\text{Nb}_4\text{MoO}_{20}\cdot 0.15 \text{H}_2\text{O}$, showing the peak assignment to $M2$ site.**
- **Fig. 4 Determination of the Mo occupancy factors in $\text{Ba}_7\text{Nb}_4\text{MoO}_{20}\cdot 0.15 \text{H}_2\text{O}$.**
- **Fig. 5 Rietveld fitting patterns of $\text{Ba}_7\text{Nb}_4\text{MoO}_{20}\cdot 0.15 \text{H}_2\text{O}$.**
- **Table 2. Refined crystal parameters and reliability factors in Rietveld analysis of the neutron diffraction data of $\text{Ba}_7\text{Nb}_4\text{MoO}_{19.849}(\text{OH})_{0.302}$ (= $\text{Ba}_7\text{Nb}_4\text{MoH}_{0.302}\text{O}_{20.151}$ = $\text{Ba}_7\text{Nb}_4\text{MoO}_{20.151}\text{H}_{0.302}$ = $\text{Ba}_7\text{Nb}_4\text{MoO}_{20}\cdot 0.151 \text{H}_2\text{O}$) at 300 K.**
- **Fig. 6 Refined crystal structure of $\text{Ba}_7\text{Nb}_4\text{MoO}_{19.849}(\text{OH})_{0.302}$ at 300 K, which shows the Mo chemical order and site occupancies of Nb and Mo atoms.**

Reviewer's comment: 3. line 89: make it clear why you are concluding that Mo order facilitates high conductivities.

Response: Thank you for your useful comment. As suggested by the reviewer, we have clarified why the chemical order of Mo atoms facilitates high conductivity as follows.

Action:

The sentences: "This work has shown that the Mo cations are localized at the $M2$ site near the ion-conducting c' layer in $\text{Ba}_7\text{Nb}_4\text{MoO}_{20}$. Thus, the Mo order would facilitate high proton and oxide-ion conductivities. The flexibility of the coordination of Nb/Mo₂ atoms near the c' layer was suggested to be a key in the high ion conduction from the *ab initio* molecular dynamics simulations³⁴. Thus, the flexibility of Mo atoms would be important for the high ion conduction in $\text{Ba}_7\text{Nb}_4\text{MoO}_{20}$ -based materials as well as other Mo-containing ionic conductors such as $\text{La}_2\text{Mo}_2\text{O}_9$ ⁴⁴."

was changed to

"The flexibility of the coordination of $M2$ atoms near the c' layer was suggested to determine the high ion conduction in $\text{Ba}_7\text{Nb}_4\text{MoO}_{20}\cdot 0.5 \text{H}_2\text{O}$ from the *ab initio* molecular dynamics simulations³⁴. Since the present work has indicated that Mo cations are localised at the $M2$ site near the ion-conducting c' layer in $\text{Ba}_7\text{Nb}_4\text{MoO}_{20}\cdot 0.15 \text{H}_2\text{O}$, the flexibility of Mo atoms is important for the high ion conduction in $\text{Ba}_7\text{Nb}_4\text{MoO}_{20}$ -based materials as well as in other Mo-containing ionic conductors such as $\text{La}_2\text{Mo}_2\text{O}_9$ ⁴⁵. Therefore, the bulk conductivity of $\text{Ba}_7\text{Nb}_{4-x}\text{Mo}_{1+x}\text{O}_{20+x/2}$ increases with increasing of the excess amount of Mo

atoms x from $x = 0$ to 0.1^{36} , which is ascribed to not only larger amount of excess oxygen atoms but also the larger amount of Mo atoms. "

Reviewer's comment: 4. There is a typo in the graphical abstract: chemical.

Response: Thank you for the comment.

Action: The spelling mistake was corrected in the revised manuscript.

Reviewer's comment: 5. Show your fitting of the ^{95}Mo NMR spectrum. Can the satellites be observed to improve precision? Models other than the $M2$ site give DFT-computed quadrupolar coupling constants close to 1 to 2 MHz, so a good fit of the experimental data is needed to confirm that the $M2$ site is the right one.

Response: Thank you for your useful comments. By fitting the ^{95}Mo peak with Lorentzian function, we were not able to determine exactly the ^{95}Mo NMR parameters, therefore, we do not show the fitting result in Fig. 3b. Thus, to examine the ^{95}Mo NMR peak position, we performed NMR spectral simulations using various quadrupolar coupling constant C_Q values and found that the peak shape observed in ^{95}Mo NMR experiments in Fig. 3b was reproduced by the simulations up to $C_Q = 2$ MHz (Fig. A2). Therefore, the experimental C_Q value is 2 MHz or less than 2 MHz.

Fig. A2 C_Q dependence of theoretical ^{95}Mo MAS (spinning frequency: 20 kHz) NMR spectra under 18.8 T calculated with (a) $(\eta, \delta_{\text{iso}}) = (0.0, 0 \text{ ppm})$ and (b) $(\eta, \delta_{\text{iso}}) = (1.0, 0 \text{ ppm})$. Apodization of 200 Hz is imposed before Fourier transformation.

A large number of satellite peaks are required to obtain precise C_Q values, as shown in Fig. A3 [Ref. 1]. However, in the present case of $\text{Ba}_7\text{Nb}_4\text{MoO}_{20}$, only four spinning side bands

were observed even in the frequency range from -1000 to 1000 ppm (Fig. A4). Therefore, we did not obtain the C_Q values using the satellite peaks.

Fig. A3 Simulated (upper) and experimental (lower) ^{95}Mo NMR spectra of the central transition of MAS samples of Cs_2MoO_4 . Reproduced from Fig. 1d in the literature [Ref. 1].

Fig. A4 ^{95}Mo NMR spectra of $\text{Ba}_7\text{Nb}_4\text{MoO}_{20}$ in a wide frequency range.

Action: The C_Q range of the peak in Table 1 was corrected to " $C_Q \leq 2$ MHz".

The description of assignment is modified as follows (yellow hatched):

"The calculated quadrupolar coupling constant $|C_Q|$ values are from 0.36 to 0.90, which also agree with the observed one (< 1 MHz)."

to

"The calculated quadrupolar coupling constant $|C_Q|$ values ranged from 0.36 to 0.90 depending on the structural model, which is consistent with the observed value (≤ 2 MHz)."

Reference

[Ref. 1] Forgeron, M. A. M.; Wasylshen, R. E. Molybdenum Magnetic Shielding and Quadrupolar Tensors for a Series of Molybdate Salts: A Solid-State ^{95}Mo NMR Study. *Phys. Chem. Chem. Phys.* **2008**, *10* (4), 574–581. <https://doi.org/10.1039/b713276j>.

Reviewer's comment: 6. Figure S5 part a and part d: please clarify shift vs shielding. I believe you are plotting shielding on the x axes but this is labelled as 'shift'.

Response: Thank you for advice to the technical term.

Action: We corrected "shift" to shielding and symbol $\delta_{\text{iso}}[\text{VASP}]$ to σ_{iso} .

Reviewer #2 (Remarks to the Author):

Reviewer's comment: This paper reports on hidden chemical order in the oxide ion conductor $\text{Ba}_7\text{Nb}_4\text{MoO}_{20}$. The authors have used a range of techniques to determine this as it is not possible via conventional neutron and/or X-ray diffraction. This new method could potentially be applied to other systems.

I am not an expert in NMR but it appears to show clear Mo order with Mo on a single site. DFT calculations back up the NMR results nicely. It is very interesting that the Mo is only found in the polyhedra where the ionic conductivity arises and it is suggested this arises due to the smaller size of Mo^{6+} compared to Mo^{5+} which is plausible.

The results are interesting and could potentially be published in Nature Communications but more insight is needed into the significance to the field and related fields.

Response: Thank you for valuable comments. One of insight into the significance to the field of solid state ionics is the importance of Mo atoms for high ion conductivity in the Mo-containing materials as discussed below in details. We have also demonstrated that the present RXRD/NMR method is powerful to examine the hidden chemical order in materials containing elements with low SCS. Thus, this work is also important for the fields of crystallography and materials science.

Action: We added the explanation of the role of Mo on ionic conduction.

"Since the present work has indicated that Mo cations are localised at the *M2* site near the ion-conducting *c'* layer in $\text{Ba}_7\text{Nb}_4\text{MoO}_{20}\cdot 0.15 \text{H}_2\text{O}$, the flexibility of Mo atoms is important for the high ion conduction in $\text{Ba}_7\text{Nb}_4\text{MoO}_{20}$ -based materials as well as in other Mo-containing ionic conductors such as $\text{La}_2\text{Mo}_2\text{O}_9$ ⁴⁵. Therefore, the bulk conductivity of $\text{Ba}_7\text{Nb}_{4-x}\text{Mo}_{1+x}\text{O}_{20+x/2}$ increases with increasing of the excess amount of Mo atoms *x* from *x* = 0 to 0.1³⁶, which is ascribed to not only larger amount of excess oxygen atoms but also the larger amount of Mo atoms."

And advantages of RXRD/NMR method were also added:

"Beyond the limits of the combined technique of conventional X-ray diffraction and NMR ("SMARTER" crystallography^{45,46}), this RXRD/NMR method can be applied to numerous compounds exhibiting chemical order/disorder of atoms with both similar X-ray atomic scattering factors and similar neutron scattering lengths (Fig. 1 and Supplementary Table 1). In contrast to the single-crystal X-ray diffraction^{19,47} and X-ray fluorescence holography⁴⁸, the RXRD/NMR method uses powders or polycrystalline samples, making it versatile and easily applicable to *in situ* measurements (e.g., at high temperatures)."

Further comments are:

Reviewer's comment: 1) The English needs checking throughout. Some of the sentences in the abstract need rewording – it's a little jumbled. There is some repetition in parts also.

Response: Thank you for the helpful comment.

Action: We used the professional English editing service Editage to correct grammatical errors and improve the wording, where the changed parts are shown in the red.

editage

Editing Certificate

This document certifies that the manuscript listed below has been edited to ensure language and grammar accuracy and is error free in these aspects. The edit was performed by professional editors at Editage, a division of Cactus Communications. The author's core research ideas were not altered in any way during the editing process. The quality of the edit has been guaranteed, with the assumption that our suggested changes have been accepted and the text has not been further altered without the knowledge of our editors.

MANUSCRIPT TITLE

Hidden chemical order in disordered Ba₇Nb₄MoO₂₀ revealed by resonant X-ray diffraction and solid-state NMR

AUTHORS

Yuta Yasui Masataka Tansho, Kotaro Fujii, Yuichi Sakuda, Atsushi Goto, Shinobu Ohki, Yuuki Mogami, Takahiro Iijima, Shintaro Kobayashi, Shogo Kawaguchi, Kellchi Osaka, Kazutaka Ikeda, Toshiya Otomo, Masatomo Yashima

ISSUED ON

February 06, 2023

JOB CODE

MLCQD_3

Vikas Narang

Vikas Narang
Chief Operating Officer - Editage

editage

Editage, a brand of Cactus Communications, offers professional English language editing and publication support services to authors engaged in over 1300 areas of research. Through its community of experienced editors, which includes doctors, engineers, published scientists, and researchers with peer review experience, Editage has successfully helped authors get published in internationally reputed journals. Authors who work with Editage are guaranteed excellent language quality and timely delivery.

GLOBAL :

+1(833) 979-0061 | request@editage.com

JAPAN :

0120-50-2987 | submissions@editage.com

CACTUS

 **impact.science**

 **researcher.life**

 **lifesciences.cactusglobal.com**

Reviewer's comment: 2) The energy barriers for oxide ion and proton conduction are higher in the “ordered” Mo system compare to the “virtual disordered”. How do you reconcile this with your results? More thought is needed here.

Response: Thank you for the valuable comment. The energy barriers for oxide ion and proton conduction are higher in the “ordered” Mo system compare to the “virtual disordered”, because bottleneck size for ion migration in the “ordered” Mo system is larger than that in the “virtual disordered system”. Therefore, we have discussed the bottleneck size.

Action:

We have described the origins of the energy barrier difference in the Discussion section as follows:

“The energy barriers for oxide-ion migration $E_{b/O}$ of Mo-ordered and virtual Mo-disordered $Ba_7Nb_4MoO_{20} \cdot 0.15 H_2O$ were calculated using the bond-valence method^{50,51}. The $E_{b/O}$ along the c axis in Mo-ordered $Ba_7Nb_4MoO_{20} \cdot 0.15 H_2O$ (1.93 eV) is higher than that in the virtual Mo-disordered $Ba_7Nb_4MoO_{20} \cdot 0.15 H_2O$ (1.60 eV) [Supplementary Table 13], which is attributable to the narrower bottleneck for oxide-ion migration along the c axis due to the higher occupancy factor of larger Nb cations at the bottleneck triangle (Supplementary Fig. 14).”

Reviewer's comment: 3) How do these results offer unprecedented insight into the understanding of the ion diffusion mechanism in hexagonal perovskite-related oxides?

Response: Thank you for the comments for improvement of the discussion. We discovered the occupational order of Mo atom at the $M2$ site adjacent to the ion-conducting layer. Therefore, an unprecedented insight is that flexibility of Mo atoms is important for the high ion conduction in $Ba_7Nb_4MoO_{20}$ -based materials.

Action:

We revised the discussion as shown below.

“Next, we discuss the influences of Mo chemical order on the material properties of $Ba_7Nb_4MoO_{20}$. The flexibility of the coordination of $M2$ atoms near the c' layer was suggested to determine the high ion conduction in $Ba_7Nb_4MoO_{20} \cdot 0.5 H_2O$ from the *ab initio* molecular dynamics simulations³⁴. Since the present work has indicated that Mo cations are localised at the $M2$ site near the ion-conducting c' layer in $Ba_7Nb_4MoO_{20} \cdot 0.15 H_2O$, the flexibility of Mo atoms is important for the high ion conduction in $Ba_7Nb_4MoO_{20}$ -based materials as well as in other Mo-containing ionic conductors such as $La_2Mo_2O_9$ ⁴⁹.”

Can you predict anything from this?

Response:

We can predict that the bulk oxide-ion conductivity of $Ba_7Nb_{4-x}Mo_{1+x}O_{20+x/2}$ increases with an increase of excess Mo content x , because the Mo atoms are suggested to have an important role in ion conduction. In fact, the bulk conductivity of $Ba_7Nb_{4-x}Mo_{1+x}O_{20+x/2}$ increases with increasing of the excess amount of Mo atoms x [Ref. 1].

[Ref. 1] Yashima, M.; Tsujiguchi, T.; Sakuda, Y.; Yasui, Y.; Zhou, Y.; Fujii, K.; Torii, S.; Kamiyama, T.; Skinner, S. J. High Oxide-Ion Conductivity through the Interstitial Oxygen Site in Ba₇Nb₄MoO₂₀-Based Hexagonal Perovskite Related Oxides. *Nat. Commun.* **2021**, *12* (1), 556. <https://doi.org/10.1038/s41467-020-20859-w>.

Would there be an optimum Nb:Mo ratio for obtaining high ionic conductivity?

Response: There would be an optimum Nb:Mo ratio for obtaining high ionic conductivity, because of (i) overdoping and (ii) impurity phase in the composition higher than the solubility limit. Optimum Nb:Cr ratio is $x = 0.15$ in Ba₇Nb_{4-x}W_xMoO_{20+x/2} [Ref. 1]. Optimum Nb:W ratio is $x = 0.20$ in Ba₇Nb_{4-x}Cr_xMoO_{20.1+x/2} [Ref. 2]. In Ba₇Ta_{4-x}Mo_{1+x}O_{20+x/2}, an optimum Ta:Mo ratio is around $x = 0.3$ due to the overdoping [Ref. 3].

- [Ref. 1] Suzuki, Y.; Murakami, T.; Fujii, K.; Hester, J. R.; Yasui, Y.; Yashima, M. Simultaneous Reduction of Proton Conductivity and Enhancement of Oxide-Ion Conductivity by Aliovalent Doping in Ba₇Nb₄MoO₂₀. *Inorg. Chem.* **2022**, *61* (19), 7537–7545. <https://doi.org/10.1021/acs.inorgchem.2c00671>.
- [Ref. 2] Sakuda, Y.; Hester, J. R.; Yashima, M. Improved Oxide-Ion and Lower Proton Conduction of Hexagonal Perovskite-Related Oxides Based on Ba₇Nb₄MoO₂₀ by Cr⁶⁺ Doping. *J. Ceram. Soc. Japan* **2022**, *130* (7), 21192. <https://doi.org/10.2109/jcersj2.21192>.
- [Ref. 3] Murakami, T.; Shibata, T.; Yasui, Y.; Fujii, K.; Hester, J. R.; Yashima, M. High Oxide-Ion Conductivity in a Hexagonal Perovskite-Related Oxide Ba₇Ta_{3.7}Mo_{1.3}O_{20.15} with Cation Site Preference and Interstitial Oxide Ions. *Small* **2022**, *18* (10), 2106785. <https://doi.org/10.1002/sml.202106785>.

What other systems could this be used for and why is it so important?

Response: Thank you for the comment.

The RXRD/NMR method (combined technique used in this study) is available for similar Ba₇Nb₄MoO₂₀-related compositions such as Ba₇Nb_{3.9}Mo_{1.1}O_{20.05} [Ref. 1], Ba₇Nb_{3.85}W_{0.15}MoO_{20.075} [Ref. 2], Ba₇Nb_{3.8}Cr_{0.2}MoO_{20.1-δ} [Ref. 3] (δ is the amount of oxygen deficiency), and Ba₃MoNbO_{8.5} [Ref. 4,5]. These materials have high oxide-ion conductivity, and oxygens diffuse two-dimensionally in the *c'* layer. However, structural analyses in these studies were performed under assumption of complete disorder of Mo atoms. Therefore, the role played by the Mo atoms remains to be elucidated. The RXRD/NMR method is effective for further understanding of ion conduction mechanism in these materials.

Action: We described possible applications of the RXRD/NMR method for other materials.

“This combined technique can be used to investigate the hidden chemical order in various ionic conductors hexagonal perovskite derivatives such as Ba₇Nb_{4-x}Mo_{1+x}O_{20+x/2}³⁶, Ba₇Nb_{4-x}W_xMoO_{20+x/2}³⁸, Ba₇Nb_{4-x}Cr_xMoO_{20.1+x/2}³⁹, and Ba₃MoNbO_{8.5}^{6,43,44} where the Mo occupancies at the *M_i* sites (*i* = 1, 2, 3, and 4) are unknown. Here *x* is the dopant or excess Mo content.”

- [Ref. 1] Yashima, M.; Tsujiguchi, T.; Sakuda, Y.; Yasui, Y.; Zhou, Y.; Fujii, K.; Torii, S.; Kamiyama, T.; Skinner, S. J. High Oxide-Ion Conductivity through the Interstitial Oxygen Site in Ba₇Nb₄MoO₂₀-Based Hexagonal Perovskite Related Oxides. *Nat. Commun.* **2021**, *12* (1), 556. <https://doi.org/10.1038/s41467-020-20859-w>.
- [Ref. 2] Suzuki, Y.; Murakami, T.; Fujii, K.; Hester, J. R.; Yasui, Y.; Yashima, M. Simultaneous Reduction of Proton Conductivity and Enhancement of Oxide-Ion Conductivity by Aliovalent Doping in Ba₇Nb₄MoO₂₀. *Inorg. Chem.* **2022**, *61* (19), 7537–7545. <https://doi.org/10.1021/acs.inorgchem.2c00671>.
- [Ref. 3] Sakuda, Y.; Hester, J. R.; Yashima, M. Improved Oxide-Ion and Lower Proton Conduction of Hexagonal Perovskite-Related Oxides Based on Ba₇Nb₄MoO₂₀ by Cr⁶⁺ Doping. *J. Ceram. Soc. Japan* **2022**, *130* (7), 21192. <https://doi.org/10.2109/jcersj2.21192>.
- [Ref. 4] Fop, S.; Skakle, J. M. S. S.; McLaughlin, A. C.; Connor, P. A.; Irvine, J. T. S. S.; Smith, R. I.; Wildman, E. J. Oxide Ion Conductivity in the Hexagonal Perovskite Derivative Ba₃MoNbO_{8.5}. *J. Am. Chem. Soc.* **2016**, *138* (51), 16764–16769. <https://doi.org/10.1021/jacs.6b10730>.
- [Ref. 5] Yashima, M.; Tsujiguchi, T.; Fujii, K.; Niwa, E.; Nishioka, S.; Hester, J. R.; Maeda, K. Direct Evidence for Two-Dimensional Oxide-Ion Diffusion in the Hexagonal Perovskite-Related Oxide Ba₃MoNbO_{8.5-δ}. *J. Mater. Chem. A* **2019**, *7* (23), 13910–13916. <https://doi.org/10.1039/C9TA03588E>.

How is this method different/better than other local structure studies?

Response: Compared to other techniques, the RXRD/NMR method used in this study has the following advantages: (i) Unlike single-crystal X-ray diffraction or X-ray fluorescence holography, it is possible to analyse powders and polycrystalline materials, as described in the introduction. (ii) It is difficult to analyse complex structures and assign peaks in the case of NMR alone or XAFS. In this study, we succeeded in analysing a complex Ba₇Nb₄MoO₂₀ containing four different *M1-M4* sites. (iii) In the experiment for polycrystalline material, high-temperature / pressure in situ experiments are easy in comparison with the experiments using single crystal because of the ability to analyse in powder or polycrystalline form.

Conventional total X-ray and neutron scattering are powerful methods, but they can not distinguish the Mo and Nb atoms from each other. If we use the total resonant X-ray scattering, we can distinguish them, which could be an interesting future work. So we have described in the text, “The novel technique would be useful for investigating not only the periodic average structure but also the short- and intermediate-range order/disorder hidden in conventional diffraction and total scattering.”

Action:

We have described in the discussion.

“Beyond the limits of the combined technique of conventional X-ray diffraction and NMR (“SMARTER” crystallography^{45,46}), this RXRD/NMR method can be applied to numerous compounds exhibiting chemical order/disorder of atoms with both similar X-ray atomic scattering factors and similar neutron scattering lengths (Fig. 1 and Supplementary Table 1). In contrast to the single-crystal X-ray diffraction^{19,47} and X-ray fluorescence

holography⁴⁸, the RXRD/NMR method uses powders or polycrystalline samples, making it versatile and easily applicable to *in situ* measurements (e.g., at high temperatures). The novel technique would be useful for investigating not only the periodic average structure but also the short- and intermediate-range order/disorder hidden in conventional diffraction and total scattering.”

Reviewer #3 (Remarks to the Author):

Reviewer's comment: The paper by Masatomo Yashima and co-authors is an interesting work on the detailed structural analysis of the high ion-conducting Ba₇Nb₄MoO₂₀, which unravels the crystallographic order/disorder of Mo and Nb in the crystal lattice. The authors used a judicious combination of advanced techniques, including high-field solid-state NMR, resonant X-ray and neutron diffractions, and DFT calculations, to determine the precise location of Mo and Nb among the four possible crystallographic sites and their occupancy rates. Although this methodology is known as SMARTER crystallography (Structure elucidation by coMbinig mAGnetic Resonance, compuTation modELing and diffRActions), (see e.g. DOI : 10.1039/C2DT30100H), it should be further developed to solve challenging structural problems such as the one presented in this manuscript. Here, the main experimental evidence comes from ⁹⁵Mo NMR and resonant X-ray diffraction, supported by DFT calculations of a number of structural models. The article is very well written. I find this work very interesting and recommend that it be accepted for publication in Nature Communications.

Response: Thank you for the useful comments. As the reviewer pointed out, this combined technique is a progress of the SMARTER crystallography.

Action: We added a description and reference for the SMARTER crystallography.

“Beyond the limits of the combined technique of conventional X-ray diffraction and NMR (“SMARTER” crystallography^{45,46}), this RXRD/NMR method can be applied to numerous compounds exhibiting chemical order/disorder of atoms with both similar X-ray atomic scattering factors and neutron scattering lengths (Fig. 1 and Supplementary Table 1).”

However, I have a few questions to clarify:

Reviewer's comment: 1. Lines 2, 33, 58, and 179: I think “resonance” X-ray diffraction is not appropriate term. It should be replaced by “resonant”. Also in Fig. captions !!

Response: Thank you for the detailed check.

Action: “resonance” has been corrected to “resonant”.

Reviewer's comment: 2. Line 83: The cell parameters are not complete (missing “b”) and do not match those in Supplementary Figures 6 & 7. This is also in Table 2, and in Supplementary Tables 9 & 10. Please check.

Response: Thank you for your comments. Supplementary Figures 6 and 7 show the local structural model where the cell parameter *a* is not equal to *b* due to the presence of Mo and Nb atoms at cation sites. Meanwhile Table 2, Supplementary Tables 9 & 10 show the

crystal parameters of the average structure refined by the Rietveld analysis. Indeed, we described in the text (Method section) as, “Atomic coordinates of $\text{Ba}_7\text{Nb}_4\text{MoO}_{20}$ were optimized in the space group $P1$, with the convergence condition of 0.02 eV \AA^{-1} .” Following the reviewer’s comment, to prevent from confusion, we added the b cell parameter as follows.

Action:

We changed from “ $a = 5.865653(4) \text{ \AA}$,” to “ $a = 5.8604(4) \text{ \AA}$ ” “ $a = b = 5.8604(4) \text{ \AA}$ ” to “ $a = b = 5.856523(4) \text{ \AA}$ ” in Supplementary Table 9, and “ $a = 5.856523(4) \text{ \AA}$ ” to “ $a = b = 5.856523(4) \text{ \AA}$ ” in Supplementary Table 10.

Reviewer's comment: 3. Fig. 1 caption: It is said, “All the SCS values are available in the Supplementary data, but we can see only some of them in the Supplementary Table 1.

Response: Thank you for the comment.

Action: We added the SCS data in the supplementary information, and as an Excel file.

Reviewer's comment: 4. Table 2: “a” and “b” are present in the endnote but missing from the Table. The same for Supplementary Table 9.

Response and Action: Thank you very much for your kind comments. We added the labels “a”, “b”, and “c” in the Table 2 and Supplementary Table 9.

Reviewer's comment: 5. Supplementary Fig. 3: The assignment of the ^{93}Nb NMR is questionable, even though is based on DFT calculations. The problem is that the observed chemical shifts do not follow the expected trend with coordination number (CN), (see doi:10.1016/j.ssnmr.2005.09.003). At first glance, I would say that the M2 site with CN 4 should correspond to the signal at -748 ppm and those at -952 and -928 ppm with CN 6 both, to M1 and M2 sites. Is it possible to integrate or calculate the area of these signals? M2 should account for 1 Nb and M2 + M3 for 4 Nb. This is likely to be difficult with the current spectrum shown in Supplementary Fig. 3 because of the overlapping sidebands with the signals. A higher magnetic field and faster MAS will improve resolution and solve this problem. Moreover, the presence of an oxide impurity is surprising for this supposedly very pure compound. An elemental analysis is strongly recommended to settle this point.

Response: Thank you for your useful comments. As suggested by the reviewer, we have examined carefully the ^{93}Nb NMR data in this work and in the literature. First, in revised Supplementary Figure 5 and Table 3, the data for NaKLaNbO_5 and Na_5NbO_5 were deleted, because these are calculated data. The data reported by Lapina et al. [Ref. 1] were also deleted, because these are out of the dotted lines in Fig. A5. The reason for the anomalous data could be that this compound containing 5-coordinated NbO_5 was not stable [Ref. 2] and decomposed before or during the measurements or the crystal structure is wrong.

Fig. A5 Correlation between experimental and calculated NMR parameters for ^{93}Nb . Na_5NbO_5 (orange square) is an outlier.

The reviewer claimed “the M2 site with CN 4 should correspond to the signal at -748 ppm and those at -952 and -928 ppm with CN 6 both, to M1 and M2 sites.” Yes, these assignments suggested by the reviewer are consistent with the trend after Lapina (Fig. A6). In our Supplementary Figure 5 and Table 3, we also used the Nb-containing oxides with NbO_4 and NbO_6 polyhedra as benchmarks. For example, LaNbO_4 contains NbO_4 and KNbO_3 contains NbO_6 . Their chemical shifts are -873 ppm for LaNbO_4 (4-coordination) and -998 ppm for KNbO_3 (6-coordination), respectively, following the trend between the chemical shift and coordination number after Lapina et al. 2005 [Ref. 1] (Fig. A6). The chemical shifts of Nb_2O_4 and Nb_1O_6 in $\text{Ba}_7\text{Nb}_4\text{MoO}_{20}$ do not follow this trend, despite the fact that the correlation between experimental and DFT calculated NMR parameters was obtained using those of LaNbO_4 and KNbO_3 , suggesting that the Nb chemical shift of $\text{Ba}_7\text{Nb}_4\text{MoO}_{20}$ is influenced not only by the coordination number, but also by other geometrical parameters and/or the electronic states of $\text{Ba}_7\text{Nb}_4\text{MoO}_{20}$. Similarly, the Mo NMR chemical shift or shielding tensor do not follow the trend with respect to the coordination number [Ref. 3, 4]. In ^{29}Si NMR spectra of zeolites, the number of the first-neighbour aluminium atoms affect to the chemical shift [Ref. 5], although this zeolite consists only SiO_4 tetrahedra.

Fig. A6. ^{93}Nb NMR chemical shift scale for NbO_y ($y = 4, 5, 6, 7,$ and 8) polyhedra. Figure reproduced from Lapina et al. (2005) [Ref. 1].

There is a correlation between the calculated and experimental C_Q values (Supplementary Figure 5e). Thus, we have assigned Nb peaks based on magnitude of C_Q values. We think that the current assignments of Nb peaks are probably correct. Therefore, we describe “Similarly, observed ^{93}Nb peaks at isotropic chemical shifts $\delta_{\text{iso}} = -748, -952$ and -928 ppm (Supplementary Table 4) can be assigned to $M1, M2$ and $M3$ sites, respectively.” It is difficult to calculate the area of peaks in the ^{93}Nb NMR spectra (Supplementary Fig. 3), because, as the reviewer point out, overlapped spinning side bands are observed. Higher magnetic fields and faster MAS could solve this problem. Furthermore, there is a problem of unassigned peak (\dagger), which makes the assignments more difficult. More advanced NMR study including the assignment is the next challenge.

Action: The NMR data of KNaLaNbO_5 and Na_5NbO_5 from the literature were removed in the revised Supplementary Fig. 5 and Table 3. The values in Supplementary Tables 3 and 4 were corrected because of new fitting without these data.

Assignments are only probable, so assignments are not shown in the revised Fig. 3a (^{93}Nb 3QMAS NMR), and Supplementary Table 4 is now labelled "possible assignments".

References

- [Ref. 1] Lapina, O. B.; Khabibulin, D. F.; Romanenko, K. V.; Gan, Z.; Zuev, M. G.; Krasil'nikov, V. N.; Fedorov, V. E. ^{93}Nb NMR Chemical Shift Scale for Niobia Systems. *Solid State Nucl. Magn. Reson.* **2005**, *28* (2–4), 204–224.
<https://doi.org/10.1016/j.ssnmr.2005.09.003>.
- [Ref. 2] Papulovskiy, E.; Shubin, A. A.; Terskikh, V. V.; Pickard, C. J.; Lapina, O. B. Theoretical and Experimental Insights into Applicability of Solid-State ^{93}Nb NMR in Catalysis. *Phys. Chem. Chem. Phys.* **2013**, *15* (14), 5115.
<https://doi.org/10.1039/c3cp44016h>.
- [Ref. 3] Tansho, M.; Goto, A.; Ohki, S.; Mogami, Y.; Sakuda, Y.; Yasui, Y.; Murakami, T.; Fujii, K.; Iijima, T.; Yashima, M. Different Local Structures of Mo and Nb Polyhedra in the Oxide-Ion-Conducting Hexagonal Perovskite-Related Oxide $\text{Ba}_3\text{MoNbO}_{8.5}$ Revealed

by ^{95}Mo and ^{93}Nb NMR Measurements. *J. Phys. Chem. C* **2022**, *126* (31), 13284–13290. <https://doi.org/10.1021/acs.jpcc.2c03429>.

[Ref. 4] Forgeron, M. A. M.; Wasylishen, R. E. A Solid-State ^{95}Mo NMR and Computational Investigation of Dodecahedral and Square Antiprismatic Octacyanomolybdate(IV) Anions: Is the Point-Charge Approximation an Accurate Probe of Local Symmetry? *J. Am. Chem. Soc.* **2006**, *128* (24), 7817–7827. <https://doi.org/10.1021/ja060124x>.

[Ref. 5] Thomas, J. M.; Fyfe, C. A.; Ramdas, S.; Klinowski, J.; Gobbi, G. C. High-Resolution Silicon-29 Nuclear Magnetic Resonance Spectrum of Zeolite ZK-4: Its Significance in Assessing Magic-Angle-Spinning Nuclear Magnetic Resonance as a Structural Tool for Aluminosilicates. *J. Phys. Chem.* **1982**, *86* (16), 3061–3064. <https://doi.org/10.1021/j100213a003>.

Moreover, the presence of an oxide impurity is surprising for this supposedly very pure compound. An elemental analysis is strongly recommended to settle this point.

Response: Thank you for your comment. The phase purity of this sample is indicated to be 100% by Rietveld analysis of X-ray diffraction and neutron diffraction data. The ICP elemental analysis shows that the cation ratio is Ba: Nb: Mo = 6.89(12): 4.078(18): 1.034(10), which is consistent with the nominal composition $\text{Ba}_7\text{Nb}_4\text{MoO}_{20}$.

The unassigned† peak in Supplementary Figure 3 is not due to impurities, but to Nb_4O_6 and/or Nb_2O_5 polyhedra in $\text{Ba}_7\text{Nb}_4\text{MoO}_{20}$.

Action: The results of elemental analysis by ICP were added to Method section as follows. “Sintered pellets were crushed and ground into fine powders to carry out X-ray powder diffraction (XRD) and TG-MS measurements.”

to

“The sintered pellets were crushed and ground into fine powders for X-ray powder diffraction (XRD), inductively coupled plasma atomic emission spectroscopy (ICP-AES, Shimadzu ICPS-8100 spectrometer), and TG-MS measurements. The ICP-AES results indicated that the cation molar ratio of $\text{Ba}_7\text{Nb}_4\text{MoO}_{20}$ was Ba: Nb: Mo = 6.89(12): 4.078(18): 1.034(10), which is consistent with the nominal composition.”

Reviewer's comment: 6. Supplementary Tables 2 & 3: In the “c” endnote, it should be “supplementary Fig. 5” (and not 6).

Response: Thank you for the careful comment.

Action: These figure numbers were corrected.

End of the response to the reviewers.

REVIEWERS' COMMENTS

Reviewer #1 (Remarks to the Author):

The authors have invested significant effort to improve the manuscript and to address the reviewers' comments. I am satisfied with the changes made and with their responses. The work should be accepted for publication.

Reviewer #2 (Remarks to the Author):

I'm happy to accept this paper now, after a couple of minor modifications as below:

1) In the abstract it is written "NMR provided direct evidence that Nb and Mo atoms occupy three and only M2 sites, respectively". Can you clarify this sentence.

2) The authors write "This combined technique can be used to investigate the hidden chemical order in various ionic conductors hexagonal perovskite derivatives such as $\text{Ba}_7\text{Nb}_4-x\text{Mo}_{1+x}\text{O}_{20+x}/236$, $\text{Ba}_7\text{Nb}_4-x\text{W}_x\text{MoO}_{20+x}/238$, $\text{Ba}_7\text{Nb}_4-x\text{Cr}_x\text{MoO}_{20.1+x}/239$, and $\text{Ba}_3\text{MoNbO}_{8.56,43,44}$ where the Mo occupancies at the M_i sites ($i = 1, 2, 3,$ and 4) are unknown. Here x is the dopant or excess Mo content."

These are all quite similar phases – can it be used more broadly?

Reviewer #3 (Remarks to the Author):

The authors have responded to all questions and points raised by the reviewers. The article can be accepted in its revised form.

RESPONSE TO REVIEWERS' COMMENTS

We are submitting the revised version of our manuscript. We appreciate the reviewers and editors for their careful reading and helpful suggestions. We have considered their feedback and incorporated their suggestions into the revised manuscript. All the changes for the action for the suggestions are highlighted by yellow and other changes are shown by the Track Changes of WORD.

We hope that our response and the revised manuscript satisfactorily address the reviewers' comments and suggestions.

Masatomo Yashima (Tokyo Institute of Technology)
on behalf of all authors

See the next pages.

Response to the Reviewer #2.

Reviewer's comment: I'm happy to accept this paper now, after a couple of minor modifications as below:

1) In the abstract it is written "NMR provided direct evidence that Nb and Mo atoms occupy three and only M2 sites, respectively". Can you clarify this sentence.

Response: Thank you for the valuable comment. If we describe the details of Nb NMR peak assignment to the M1, M2 and M3 sites, we can clarify this sentence. But this would violate the word limit. As the Nb NMR assignment is not very important, we have described only the Mo NMR results for clarification.

Action: We have changed from

"NMR provided direct evidence that Nb and Mo atoms occupy three and only M2 sites, respectively, where the M2 site is located near the intrinsically oxygen-deficient ion-conducting layer."

to

"NMR provided direct evidence that Mo atoms occupy only the M2 site near the intrinsically oxygen-deficient ion-conducting layer."

Reviewer's comment: 2) The authors write "This combined technique can be used to investigate the hidden chemical order in various ionic conductors hexagonal perovskite derivatives such as $Ba_7Nb_{4-x}Mo_{1+x}O_{20+x}/236$, $Ba_7Nb_{4-x}W_xMoO_{20+x}/238$, $Ba_7Nb_{4-x}Cr_xMoO_{20.1+x}/239$, and $Ba_3MoNbO_{8.56,43,44}$ where the Mo occupancies at the M_i sites (i = 1, 2, 3, and 4) are unknown. Here x is the dopant or excess Mo content." These are all quite similar phases – can it be used more broadly?

Response: Thank you for the useful comment. As suggested by the reviewer, we have added two examples ($Ag_{1-x}Cd_xSbTe_2$ and Zr_5Ir_2Os) in the text.

Action: We changed from

"Beyond the limits of the combined technique of conventional X-ray diffraction and NMR ("SMARTER" crystallography^{45,46}), this RXRD/NMR method can be applied to numerous compounds exhibiting chemical order/disorder of atoms with both similar X-ray atomic scattering factors and similar neutron scattering lengths (Fig. 1 and Supplementary Table 1)."

to

"Beyond the limits of the combined technique of conventional X-ray diffraction and NMR ("SMARTER" crystallography^{45,46}), this RXRD/NMR method can be applied to numerous compounds such as thermoelectric $Ag_{1-x}Cd_xSbTe_2$ ¹⁶ and superconducting Zr_5Ir_2Os ⁴⁷ exhibiting chemical order/disorder of atoms with both similar X-ray atomic scattering factors and similar neutron scattering lengths (Fig. 1 and Supplementary Table 1)."

End of the response to the reviewers.